# When and why does motor preparation arise in recurrent neural network models of motor control?

Marine Schimel[1]*, Ta-Chu Kao[2], Guillaume Hennequin[1]

[1]Computational and Biological Learning Lab, Department of Engineering, University of Cambridge, Cambridge, United Kingdom; [2]Meta Reality Labs, Burlingame, United States

**Abstract** During delayed ballistic reaches, motor areas consistently display movement-specific activity patterns prior to movement onset. It is unclear why these patterns arise: while they have been proposed to seed an initial neural state from which the movement unfolds, recent experiments have uncovered the presence and necessity of ongoing inputs during movement, which may lessen the need for careful initialization. Here, we modeled the motor cortex as an input-driven dynamical system, and we asked what the optimal way to control this system to perform fast delayed reaches is. We find that delay-period inputs consistently arise in an optimally controlled model of M1. By studying a variety of network architectures, we could dissect and predict the situations in which it is beneficial for a network to prepare. Finally, we show that optimal input-driven control of neural dynamics gives rise to multiple phases of preparation during reach sequences, providing a novel explanation for experimentally observed features of monkey M1 activity in double reaching.

*For correspondence:
mmcs3@cam.ac.uk

Competing interest: The authors declare that no competing interests exist.

## eLife assessment

This **important** study provides a new perspective on why preparatory activity occurs before the onset of movement. The authors report that when there is a cost on the inputs, the optimal inputs should start before the desired network output for a wide variety of recurrent networks. The authors present **compelling** evidence by combining mathematically tractable analyses in linear networks and numerical simulation in nonlinear networks.

## Introduction

During the production of ballistic movements, the motor cortex is thought to operate as a dynamical system whose state trajectories trace out the appropriate motor commands for downstream effectors (*Shenoy et al., 2013*; *Miri et al., 2017*; *Russo et al., 2018*). The extent to which these cortical dynamics are controlled by exogenous inputs before and/or during movement is the subject of ongoing study.

On the one hand, several experimental and modeling studies point to a potential role for exogenous inputs in motor preparation. First, cortical state trajectories are empirically well described by a low-dimensional dynamical system evolving near-autonomously during movement (*Churchland et al., 2012*; *Pandarinath et al., 2018*; *Schimel et al., 2022*), such that there is a priori no reason to suspect that inputs are required for motor production. Rather, inputs would be required during preparation to bring the state of the cortical network into a suitable initial condition. This input-driven seeding process is corroborated by observations of movement-specific primary motor cortex (M1) activity arising well before movement initiation (*Lara et al., 2018*; *Kaufman et al., 2014*; *Churchland et al.,*

**Figure 1.** Control is possible under different strategies. (**A**) Trial-averaged firing rate of two representative monkey primary motor cortex (M1) neurons, across eight different movements, separately aligned to target onset (left) and movement onset (right). Neural activity starts separating across movements well before the animal starts moving. (**B**) Top: a recurrent neural network (RNN) model of M1 dynamics receives external inputs $\boldsymbol{u}(t)$ from a higher-level controller, and outputs control signals for a biophysical two-jointed arm model. Inputs are optimized for the correct production of eight center-out reaches to targets regularly positioned around a circle. Bottom: firing rate of a representative neuron in the RNN model for each reach, under two extreme control strategies. In the first strategy (left, solid lines), the external inputs $\boldsymbol{u}(t)$ are optimized while being temporally confined to the preparatory period. In the second strategy (right, dashed lines), they are optimized while confined to the movement period. Although slight differences in hand kinematics can be seen (compare corresponding solid and dashed hand trajectories), both control policies lead to successful reaches. These introductory simulations are shown for illustration purposes; the particular choice of network connectivity and the way the control inputs were found are described in the Results section.

*2012*; *Meirhaeghe et al., 2023*; *Figure 1A*), and associated models demonstrate the critical role of preparatory inputs therein (*Sussillo et al., 2015*; *Hennequin et al., 2014*; *Kao et al., 2021*).

On the other hand, recent studies in mice have shown that the motor cortex receives critical pattern-generating input from the thalamus during movement production (*Sauerbrei et al., 2020*), and recurrent neural network (RNN)-based modeling of the motor feedback loop involved in reaching movements suggests that sensory feedback may also contribute significantly to the observed dynamics of M1 (*Kalidindi et al., 2021*). Moreover, most published network models of delayed reaches are able to perform the task just as well without preparatory inputs, i.e., with external inputs forcefully confined to the movement epoch – an illustratory example is shown in *Figure 1B*. Thus, the relative contributions of preparatory vs. movement-epoch inputs to the dynamics implemented by M1 (potentially as part of a broader set of areas) remain unclear.

In addition to the specific form that inputs to cortical dynamics might take, one may ask more broadly about the computational role of motor preparation. Motor preparation is known to benefit behavior (e.g. by shortening reaction times and enabling more accurate execution *Riehle and Requin, 1989*; *Churchland and Shenoy, 2007*; *Michaels et al., 2015*) and may facilitate motor learning (*Sheahan et al., 2016*; *Sun et al., 2022*). However, from the perspective of cortical dynamics, preparation also introduces additional constraints.

Specifically, the high density of M1 neurons projecting directly to the spinal cord (*Dum and Strick, 1991*) suggests that motor cortical outputs control lower-level effectors with little intermediate processing. For preparatory processes to avoid triggering premature movement, any pre-movement activity in the motor and dorsal premotor (PMd) cortices must therefore engage the pyramidal tract neurons in a way that ensures their activity patterns will not lead to any movement.

While this can be achieved by constraining neural activity to evolve in a nullspace of the motor output (*Kaufman et al., 2014*), the question nevertheless arises: what advantage is there to having neural dynamics begin earlier in a constrained manner, rather than unfold freely just in time for movement production?

Here, we sought a normative explanation for motor preparation at the level of motor cortex dynamics: we asked whether preparation arises in RNNs performing delayed-reaching tasks, and what factors lead to more or less preparation.

Such an explanation could not be obtained from previous network models of delayed reaches, as they typically assume from the get-go that the cortical network receives preparatory inputs during a fixed time window preceding the go cue (*Sussillo et al., 2015*; *Kao et al., 2021*). In this case, pre-movement activity is by designing a critical determinant of the subsequent behavior (*Sussillo et al., 2015*; *Kao et al., 2021*; *Zimnik and Churchland, 2021*). In this work, we removed this modeling assumption and studied models in which the correct behavior could in principle be obtained without explicit motor preparation.

To study the role of motor preparation, and that of exogenous inputs in this process, we followed an optimal control approach (*Harris and Wolpert, 1998*; *Todorov and Jordan, 2002*; *Yeo et al., 2016*). We considered the dynamics of an RNN model of M1 coupled to a model arm (*Todorov and Li, 2003*), and used a standard control cost functional to quantify and optimize performance in a delayed-reaching task. We used the iterative linear quadratic regulator algorithm (iLQR) algorithm (*Li and Todorov, 2004*) to find the spatiotemporal patterns of network inputs that minimize this cost functional, for any given network connectivity. Critically, these inputs could arise both before and during movement; thus, our framework allowed for principled selection among a continuum of motor strategies, going from purely autonomous motor generation following preparation, to purely input-driven unprepared dynamics.

We considered an inhibition-stabilized network – which was shown previously to capture prominent aspects of monkey M1 activity (*Hennequin et al., 2014*; *Kao et al., 2021*) – and found that optimal control of the model requires preparation, with optimal inputs arising well before movement begins. To understand what features of network connectivity lead to optimal preparatory control strategies, we first turned to low-dimensional models, which could be more easily dissected. We then generalized insights from those systems back to high-dimensional networks using tools from control theory, and found that preparation can be largely explained by two quantities summarizing the dynamical response properties of the network.

Finally, we studied the optimal control of movement *sequences*. Consistent with recent experimental findings (*Zimnik and Churchland, 2021*), we observed that optimal control of compound reaches leads to input-driven preparatory activity in a dedicated activity subspace prior to each movement.

Overall, our results show that preparatory neural activity patterns arise from optimal control of reaching movements at the level of motor cortical circuits, thus providing a possible explanation for a number of observed experimental findings.

## Model

### A model of cortical dynamics for reaching movements

We considered a simple reaching task, in which the hand must move from a resting location to one of eight radially located targets in a 2D plane as fast as possible (*Figure 1*). The target had to be reached within 600 ms of a go cue that follows a delay period of varying (but known) duration. We modeled the trajectory of the hand via a two-jointed model arm (*Li and Todorov, 2004*; *Kao et al., 2021*), driven into motion by a pair of torques $\boldsymbol{m}(t)$ (Methods). We further assumed that these torques arise as a linear readout of the momentary firing rates $\boldsymbol{r}(t)$ of a population of M1 neurons,

$$\boldsymbol{m}(t) = \boldsymbol{C}\boldsymbol{r}(t), \tag{1}$$

where $\boldsymbol{C}$ was a randomly generated readout matrix, projecting the neural activity into the output space. We modeled the dynamics of $N = 200$ M1 neurons using a standard rate equation,

$$\tau \frac{d\boldsymbol{x}(t)}{dt} = -\boldsymbol{x}(t) + \boldsymbol{W}\boldsymbol{r}(t) + \boldsymbol{h} + \boldsymbol{u}(t) \tag{2}$$

$$r(t) = \phi \left[ \boldsymbol{x}(t) \right], \tag{3}$$

where the momentary population firing rate vector $\boldsymbol{r}(t)$ was obtained by passing a vector of internal neuronal activations $\boldsymbol{x}(t)$ through a rectified linear function $\phi \left[ \cdot \right]$, element-wise. In *Equation 2*, $\boldsymbol{h}$ is a constant input that establishes a baseline firing rate of 5 Hz on average, with a standard deviation of 5 Hz across neurons, $\boldsymbol{u}(t)$ is a task-dependent control input (see below), and $\boldsymbol{W}$ denotes the matrix of recurrent connection weights. Throughout most of this work, we considered inhibition-stabilized M1 dynamics (*Hennequin et al., 2014*; Methods), which have previously been shown to produce activity resembling that of M1 during reaching (*Kao et al., 2021*).

Thus, our model can be viewed as a two-level controller, with the arm being controlled by M1, and M1 being controlled by external inputs. Note that each instantiation of our model corresponds to a set of $\boldsymbol{W}$, $\boldsymbol{C}$, and $\boldsymbol{h}$, none of which are specifically optimized for the task.

## To prepare or not to prepare?

Previous experimental (*Churchland et al., 2012*; *Shenoy et al., 2013*) and modeling (*Hennequin et al., 2014*; *Sussillo et al., 2015*; *Pandarinath et al., 2018*) work suggests that fast ballistic movements rely on strong dynamics, which are observed in M1 and well modeled as near-autonomous (although the underlying dynamical system may not be anatomically confined to M1, as we discuss later). Network-level models of ballistic control thus rely critically on a preparation phase during which they are driven into a movement-specific state that seeds their subsequent autonomous dynamics (*Kao et al., 2021*; *Sussillo et al., 2015*). However, somewhat paradoxically, the same recurrent neural network models can also solve the task in a completely different regime, in which task-related inputs arise during movement only, with no preparatory inputs whatsoever. We illustrate this dichotomy in *Figure 1*. The same center-out reach can be produced with control inputs to M1 that arise either prior to movement only (full lines), or during movement only (dashed lines). In the latter case, no reach-specific preparatory activity is observed, making the model inconsistent with experimental findings. But what rationale is there in preparing for upcoming movements, then?

To address this question, we formulated delayed reaching as an optimal control problem, and asked what external inputs are required, and at what time, to drive the hand into the desired position with minimum control effort. Specifically, we sought inputs that were as weak as possible yet accurately drove the hand to the target within an allotted time window. We also penalized inputs that caused premature movement before the go cue.

Thus, we solved for spatiotemporal input trajectories that minimized a cost functional capturing the various task requirements. Our cost was composed of three terms: $\mathcal{J}_{\text{target}}$ penalizes deviations away from the target, with an 'urgency' weight that increases quadratically with time, thus capturing the implicit incentive for animals to perform fast reaches in such experiments (which are normally conducted in sessions of fixed duration).

$\mathcal{J}_{\text{null}}$ penalizes premature movement during preparation, as measured by any deviation in position, speed, and acceleration of the hand. Finally, $\mathcal{J}_{\text{effort}}$ penalizes control effort in the form of input magnitude throughout the whole trial, thus promoting energy-efficient control solutions among a typically infinite set of possibilities (*Kao et al., 2021*; *Sterling and Laughlin, 2015*). Note that $\mathcal{J}_{\text{effort}}$ can be viewed as a standard regularization term, and must be included to ensure the control problem is well defined. The total objective thus had the following form:

$$\mathcal{J} \left[ \boldsymbol{u}(t) \right] = \underbrace{\int_0^T \| \boldsymbol{\theta}(t) - \boldsymbol{\theta}^\star \|^2 \frac{t^2}{T^2} \frac{dt}{T}}_{\mathcal{J}_{\text{target}}}$$

$$+ \alpha_{\text{null}} \underbrace{\int_{-\Delta_{\text{prep}}}^0 \left( \| \boldsymbol{\theta}(t) - \boldsymbol{\theta}_0 \|^2 + \| \dot{\boldsymbol{\theta}}(t) \|^2 + \| \boldsymbol{m}(t) \|^2 \right) \frac{dt}{T}}_{\mathcal{J}_{\text{null}}} \tag{4}$$

$$+ \alpha_{\text{effort}} \underbrace{\int_{-\Delta_{\text{prep}}}^T \| \boldsymbol{u}(t) \|^2 \frac{dt}{NT}}_{\mathcal{J}_{\text{effort}}},$$

where $\boldsymbol{\theta}$ and $\dot{\boldsymbol{\theta}}$ denote the position and velocity of the hand in angular space, $\Delta_{\mathrm{prep}}$ was the duration of the delay period, and $T$ that of the movement period. As $\mathcal{J}_{\mathrm{target}}$ and $\mathcal{J}_{\mathrm{null}}$ depend on $\boldsymbol{u}(t)$ implicitly through *Equations 1 and 2*, $\mathcal{J}$ is a function of $\boldsymbol{u}$ only.

Importantly, we allowed for inputs within a time window beginning $\Delta_{\mathrm{prep}}$ ms before, and ending $T$ ms after the go cue (set at $t = 0$). Therefore, both preparation-only and movement-only input strategies (*Figure 1*) could potentially arise, as well as anything in-between.

Note that this control objective assumes that the delay duration ($\Delta_{\mathrm{prep}}$) is known ahead of time, an assumption that does not hold for many delayed-reaching tasks in monkeys where the delay is uncertain. We make this assumption for computational tractability and later discuss extensions to the uncertain case (Discussion).

Here, we solved for the optimal control inputs using the iLQR (*Li and Todorov, 2004*), an efficient trajectory optimization algorithm that is well suited for handling the nonlinear nature of both the arm's and the network's dynamics. As our primary goal was to assess the role of preparation in a normative way, we did not study the putative circuit dynamics upstream of M1 that might lead to the computation of these optimal inputs.

We balanced the various components of our cost functional by choosing $\alpha_{\mathrm{null}}$ and $\alpha_{\mathrm{effort}}$ to qualitatively match the behavioral requirements of a typical reach-and-hold task. Specifically, we tuned them jointly so as to ensure (i) stillness during preparation and (ii) reach duration of approximately ~400 ms, with the hand staying within 0.5 cm of the target for ~200 ms after the end of the reach. We ensured that the main qualitative features of the solution, i.e., the results presented below, were robust to the choice of hyperparameter values within the fairly large range in which the above soft-constraints are satisfied (Appendix 1).

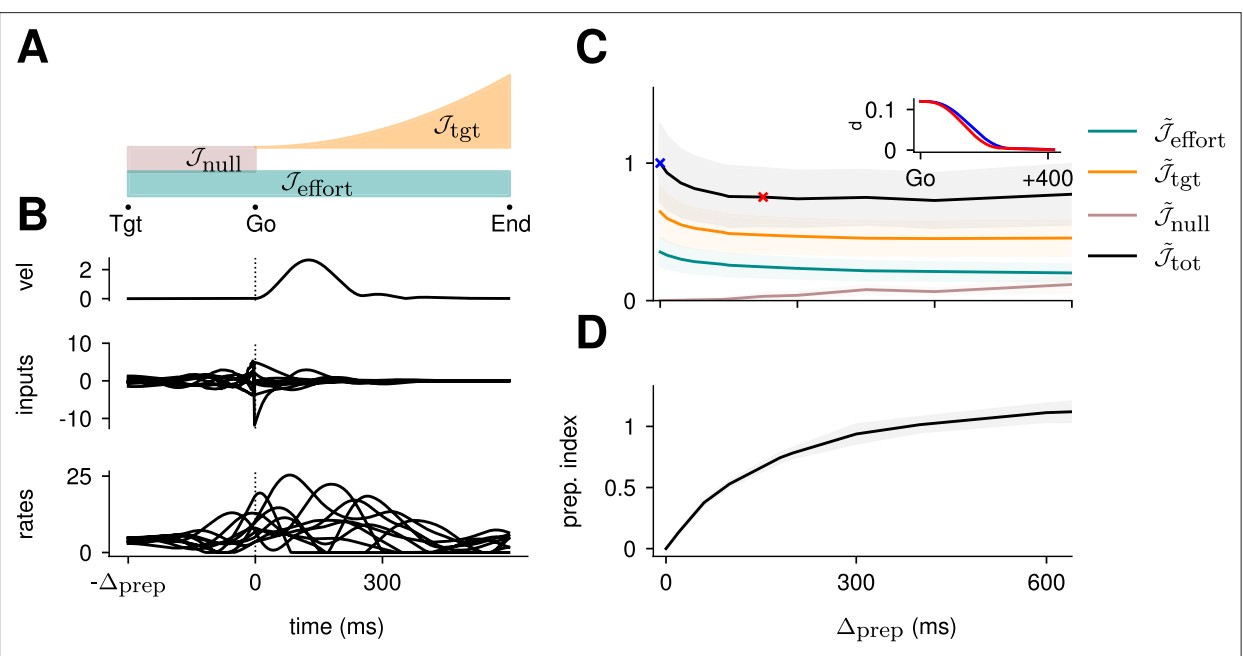

**Figure 2.** Optimal control of the inhibition-stabilized network (ISN). (**A**) Illustration of the different terms in the control cost function, designed to capture the different requirements of the task. 'Tgt' marks the time of target onset, 'Go' that of the go cue (known in advance), and 'End' the end of the trial. (**B**) Time course of the hand velocity (top), optimal control inputs (middle; 10 example neurons), and firing rates (bottom, same neurons) during a delayed reach to one of the eight targets shown in *Figure 1A*. Here, the delay period was set to $\Delta_{\mathrm{prep}} = 300$ ms. Note that inputs arise well before the go cue, even though they have no direct effect on behavior at that stage. (**C**) Dependence of the different terms of the cost function on preparation time. All costs are normalized by the total cost at $\Delta_{\mathrm{prep}} = 0$ ms. The inset shows the time course of the hand's average distance to the relevant target when no preparation is allowed (blue) and when preparation is allowed (red). Although the target is eventually reached for all values of $\Delta_{\mathrm{prep}}$, the hand gets there faster with longer preparation times, causing a decrease in $\mathcal{J}_{\mathrm{tgt}}$ – and therefore also in $\mathcal{J}_{\mathrm{tot}}$. Another part of the decrease in $\mathcal{J}_{\mathrm{tot}}$ is due to a progressively lower input energy cost $\mathcal{J}_{\mathrm{effort}}$. On the other hand, the cost of staying still before the go cue increases slightly with $\Delta_{\mathrm{prep}}$. (**D**) We define the preparation index as the ratio of the norms of the external inputs during preparation and during movement (see text). The preparation index measures how much the optimal strategy relies on the preparatory period. As more preparation time is allowed, this is used by the optimal controller and more inputs are given during preparation. For longer preparation times, this ratio increases sub-linearly, and eventually settles.

## Results

### Preparation arises as an optimal control strategy

Using the above control framework, we assessed whether the optimal way of performing a delayed reach involves preparation.

More concretely, does the optimal control strategy of the model described in *Equation 2* involve any preparatory inputs during the delay period? For any single optimally performed reach, we found that network activity began changing well before the go cue (*Figure 2B*, bottom), and that this was driven by inputs that arose early (*Figure 2B*, middle). Thus, although preparatory network activity cancels in the readout (such that the hand remains still; *Figure 2B*, top) and therefore does not contribute directly to movement, it still forms an integral part of the optimal reach strategy.

To quantify how much the optimal control strategy relied on inputs prior to movement, we defined the *preparation index* as the ratio of input magnitude during the delay period to that during the remainder of the trial:

$$\text{prep. index} = \sqrt{\frac{\int_{-\Delta_{\text{prep}}}^{0} \|\boldsymbol{u}(t)\|^2 dt}{\int_{0}^{T} \|\boldsymbol{u}(t)\|^2 dt}}. \tag{5}$$

We found that the preparation index rose sharply as we increased the delay period, and eventually plateaued at ~1.3 for delay periods longer than 300 ms (*Figure 2C*).

Similarly, the total cost of the task was highest in the absence of preparation, and decreased until it also reached a plateau at $\Delta_{\text{prep}} \sim 300$ ms (*Figure 2C*, black). This appears somewhat counterintuitive, as having a larger $\Delta_{\text{prep}}$ means that both $\mathcal{J}_{\text{effort}}$ and $\mathcal{J}_{\text{null}}$ are accumulated over a longer period. To resolve this paradox, we examined each component of the cost function. We found that the overall decrease in cost with increasing preparation time was driven by a concurrent decrease in both $\mathcal{J}_{\text{tgt}}$ and $\mathcal{J}_{\text{effort}}$. The former effect was due to the model producing faster reaches (*Figure 2C* inset; hand position for a reach with [red] and without [blue] preparation) while the latter arose from smaller control inputs being necessary when preparation was allowed. Together, these results suggest that the presence of a delay period changes the optimal control strategy for reaching, and increases performance in the task.

The results above show that delaying the reach beyond ~300 ms brings little benefit; in particular, all components of the cost stabilize past that point (*Figure 2C*). We thus wondered what features the

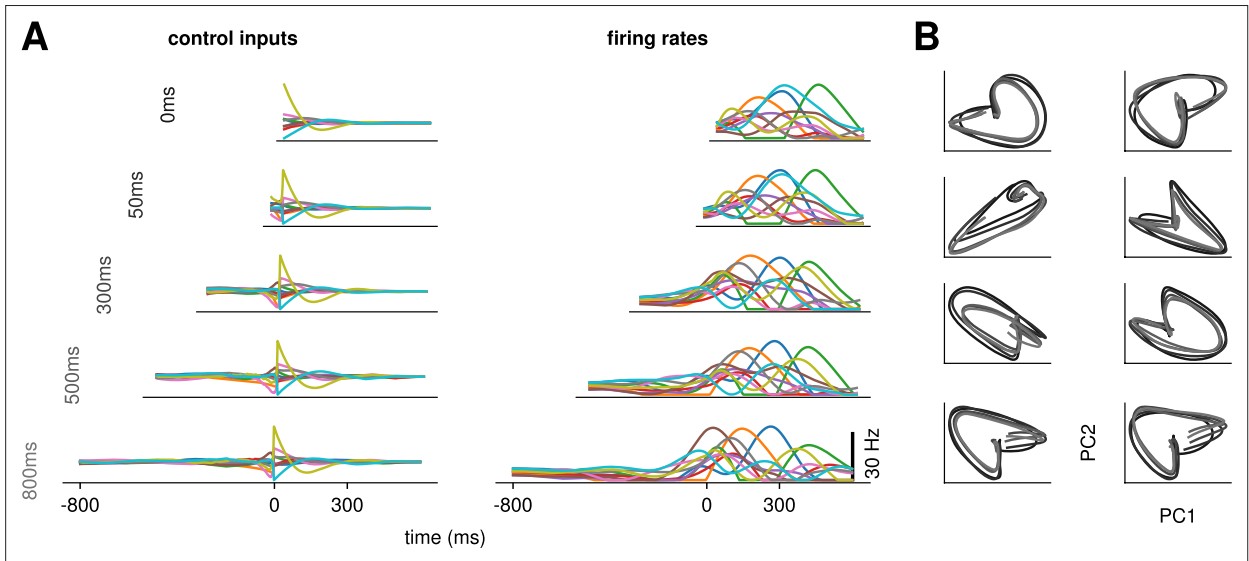

**Figure 3.** Conservation of the optimal control strategy across delays. (**A**) Optimal control inputs to 10 randomly chosen neurons in the model recurrent neural network (RNN) (left) and their corresponding firing rates (right) for different preparation times $\Delta_{\text{prep}}$ (ranging from 0 to 800 ms; c.f. labels). (**B**) Projection of the movement-epoch population activity for each of the eight reaches (panels) and each value of $\Delta_{\text{prep}}$ shown in A (darker to lighter colors). These population trajectories are broadly conserved across delay times, and become more similar for larger delays.

optimally controlled dynamics would display as $\Delta_{\text{prep}}$ increased beyond 300 ms. Would the network defer preparation to a last minute surge, or prepare more gently over the entire preparatory window?

Would the network produce the same neural activity patterns? We found that the optimal controller made very little use of any preparation time available up to 300 ms before the go cue: with longer preparation times, external input continued to arise just a couple of hundred milliseconds before movement initiation, and single neuron firing rates remained remarkably similar (*Figure 3A*). This was also seen in PCA projections of the firing rates, which traced out similar trajectories irrespective of the delay period (*Figure 3B*). We hypothesized that this behavior is due to the network dynamics having a certain maximum characteristic timescale, such that inputs that arrive too early end up being 'forgotten' – they increase $\mathcal{J}_{\text{effort}}$ and possibly $\mathcal{J}_{\text{null}}$ without having a chance to influence $\mathcal{J}_{\text{tgt}}$. We confirmed this by varying the characteristic time constant ($\tau$ in *Equation 2*). For a fixed $\Delta_{\text{prep}}$, we found that for larger (resp. lower) values of $\tau$, the optimal control inputs started rising earlier (resp. later) and thus occupied more (resp. less) of the alloted preparatory period (*Appendix 1—figure 3*).

## Understanding optimal control in simplified models

Having established that the inhibition-stabilized network (ISN) model of M1 relies on preparatory inputs to solve the delayed-reaching task, we next tried to understand *why* it does so.

To further unravel the interplay between the structure of the network and the optimal control strategy, i.e., what aspects of the dynamics of the network warrant preparation, we turned to simpler, two-dimensional (2D) models of cortical dynamics. These 2D models are small enough to enable detailed analysis (*Appendix 1—figure 2*), yet rich enough to capture the two dominant dynamical phenomena that arise in ISN dynamics: nonnormal amplification (*Murphy and Miller, 2009*; *Goldman, 2009*; *Hennequin, 2012*) and oscillations (*Brunel, 2000*; *Dayan and Abbott, 2001*). Specifically, networks of E and I neurons have been shown to embed two main motifs of effective connectivity which are revealed by appropriate orthogonal changes of basis: (i) feedforward ('nonnormal') connectivity whereby a 'source mode' of E-I imbalance feeds into a 'sink mode' in which balance is restored, and (ii) anti-symmetric connectivity that causes the two populations to oscillate.

To study the impact of each of these prototypical connectivity motifs on movement preparation, we implemented them separately, i.e., as two small networks of two units each, with an overall connectivity scale parameter $w$ which we varied (*Figure 4A and D*; Methods). As both nonnormal and oscillatory dynamics arise from linear algebraic properties of the connectivity matrix, we considered linear network dynamics for this analysis ($\phi(x) = x$ in *Equation 3*). Moreover, to preserve the existence of an output nullspace in which preparation could in principle occur without causing premature movement, we reduced the dimensionality of the motor readout from 2D (where there would be no room left for a nullspace) to 1D (leaving a 1D nullspace), and adapted the motor task so that the network now had to move the hand position along a single dimension (*Figure 4B and E*, top). Analogous to the previous arm model, we assumed that the hand's acceleration along this axis was directly given by the 1D network readout.

We found that optimal control of both dynamical motifs generally led to preparatory dynamics, with inputs arising before the go cue (*Figure 4B and E*, bottom). In the feedforward motif, the amount of preparatory inputs appeared to depend critically on the orientation of the readout. When the readout was aligned with the sink (brown) mode (*Figure 4B*, left), the controller prepared the network by moving its activity along the source (orange) mode (*Figure 4C*, left). This placed the network in a position from which it had a natural propensity to generate large activity transients along the readout dimension (c.f. flow field in *Figure 4A*); here, these transients were exploited to drive the fast upstroke in hand acceleration and throw the hand toward the target location. Note that this strategy reduces the amount of input the controller needs to deliver during the movement, because the network itself does most of the work.

Nevertheless, in this case the network's own impulse response was not rich enough to accommodate the phase reversal required to subsequently slow the hand down and terminate the movement. Optimal control therefore also involved inputs during the movement epoch, leading to a preparatory index of ~0.54 (*Figure 4G*, dark blue).

When it was instead the source mode that was read out (*Figure 4B*, right), the only dimension along which the system could prepare without moving was the sink mode. Preparing this way is of

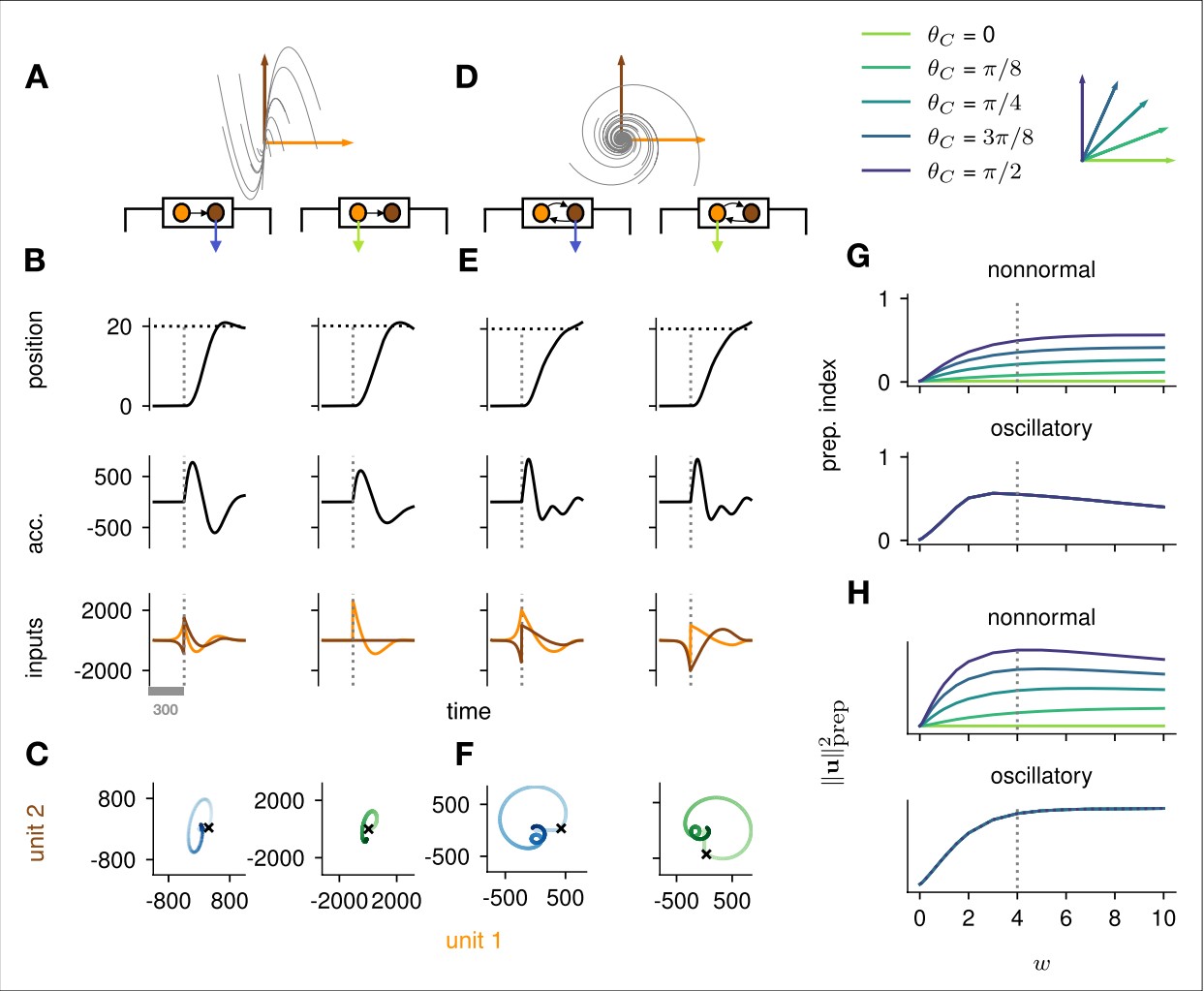

**Figure 4.** Analysis of the interplay between the optimal control strategy and two canonical motifs of E-I network dynamics: nonnormal transients driven by feedforward connectivity (**A–C**), and oscillations driven by anti-symmetric connectivity (**D–F**). (**A**) Activity flow field (10 example trajectories) of the nonnormal network, in which a 'source' unit (orange) drives a 'sink' unit (brown). We consider two opposite readout configurations, where it is either the sink (left) or the source (right) that drives the acceleration of the hand. (**B**) Temporal evolution of the hand position (top; the dashed horizontal line indicates the reach target), hand acceleration (middle), and optimal control inputs to the two units (bottom; colors matching panel A), under optimal control given each of the two readout configurations shown in A (left vs. right). The dashed vertical line marks the go cue, and the gray bar indicates the delay period. While the task can be solved successfully in both cases, preparatory inputs are only useful when the sink is read out. (**C**) Network activity trajectories under optimal control. Each trajectory begins at the origin, and the end of the delay period is shown with a black cross. (**D–F**) Same as (**A–C**), for the oscillatory network. (**G–H**) Preparation index (top) and total amount of preparatory inputs (bottom) as a function of the scale parameter $w$ of the network connectivity, for various readout configurations (color-coded as shown in the top inset). The nonnormal network (top) prepares more when the readout is aligned to the most controllable mode, while the amount of preparation in the oscillatory network (bottom) is independent of the readout direction. The optimal strategy must balance the benefits from preparatory inputs which allow to exploit the intrinsic network dynamics, with the constraint to remain still. This is more difficult when the network dynamics are strong and pushing activity out of the readout-null subspace, explaining the decrease in preparation index for large values of $w$ in the oscillatory network.

no benefit, because the flow field along the sink mode has no component along the source (readout) mode.

Thus, here the optimal strategy was to defer control to the movement epoch, during which the transient growth of network activity along the readout rested entirely on adequate control inputs. This led to a preparation index of ~0 (*Figure 4G*, pale green). Although the network did react with large activity excursions along the sink mode (*Figure 4C*, right), these were inconsequential for the movement. Importantly, of the two extreme readout configurations discussed above, the first one yielded a smaller overall optimal control cost (by a factor of ~1.5). Thus, at a meta-control level,

ideal downstream effectors would read out the sink mode, not the source mode. Note that while increasing the connectivity strength initially led to more preparation (*Figure 4H*), a plateau was eventually reached for $w \geq 4$. Indeed, while stronger dynamics initially make preparation more beneficial, they also make it more difficult for preparatory activity to remain in the readout nullspace.

We obtained similar insights for oscillatory network dynamics (*Figure 4D–F*). A key difference however was that the flow field was rotationally symmetric such that no distinction could be made between 'source' and 'sink' units – indeed the optimal control strategy yielded the same results (up to a rotation of the state space) irrespective of which of the two units was driving the hand's acceleration (compare left and right panels in *Figure 4D–F*). Nevertheless, the optimal controller consistently moved the network's activity along the output-null axis during preparation, in such a way as to engage the network's own rotational flow immediately after the go cue (*Figure 4F*). This rotational flow drove a fast rise and decay of activity in the readout unit, thus providing the initial segment of the required hand acceleration. The hand was subsequently slowed down by modest movement-epoch control inputs which eventually receded, leading to a preparation index of ~0.58. Interestingly, the preparation index showed a decrease for very large $w$ (*Figure 4G*), which did not reflect smaller preparatory inputs (*Figure 4H*) but rather reflected the larger inputs that were required during movement to cancel the fast oscillations naturally generated by the network.

The above results highlight how the optimal control strategy is shaped by the dynamical motifs present in the network. Crucially, we found that the optimal way to control the movement depends not only on the strength and flow of the internal network dynamics, but also on their interactions with the readout.

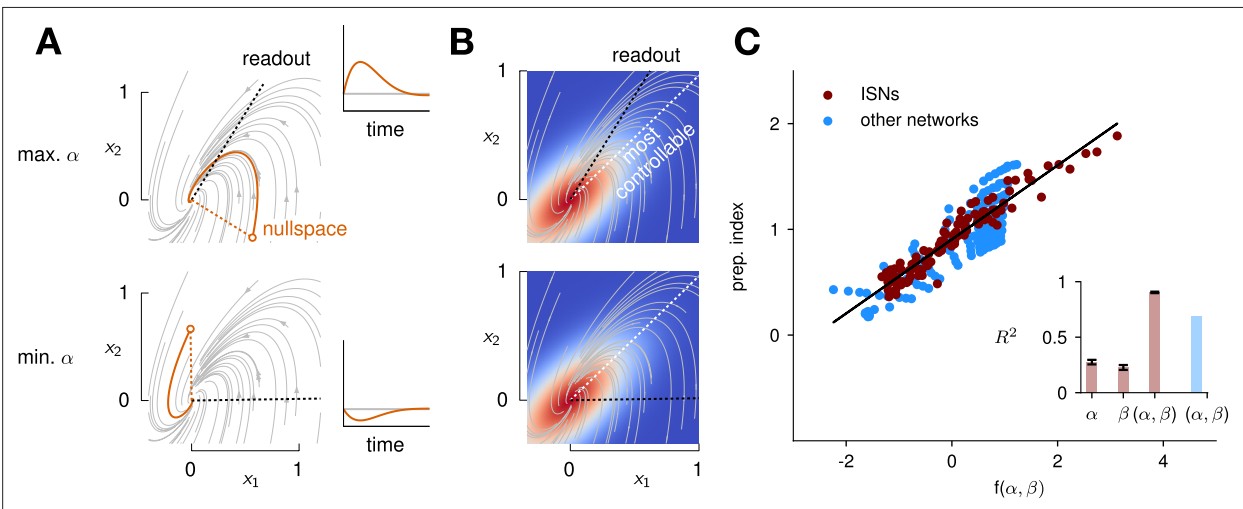

**Figure 5.** Predicting the preparation index from the observability of the output nullspace ($\alpha$) and the controllability of the readout ($\beta$, see details in text). (**A**) Illustration of the observability of the output nullspace in a synthetic two-dimensional system. The observability of a direction is characterized by how much activity (integrated squared norm) is generated along the readout by a unit-norm initial condition aligned with that direction. The top and bottom panels show the choices of readout directions (dotted black) for which the corresponding nullspace (dotted orange) is most (maximum $\alpha$) and least (minimum $\alpha$) observable, respectively. Trajectories initialized along the null direction are shown in solid orange, and their projections onto the readout are shown in the inset. (**B**) Illustration of the controllability of the readout in the same 2D system as in (**A**). To compute controllability, the distribution of activity patterns collected along randomly initialized trajectories is estimated (heatmap); the controllability of a given direction then corresponds to how much variance it captures in this distribution. Here, the network has a natural propensity to generate activity patterns aligned with the dashed white line ('most controllable' direction). The readout directions are repeated from panel A (dotted black). The largest (resp. smallest) value of $\beta$ would by definition be obtained when the readout is most (resp. least) controllable. Note the tradeoff in this example: the choice of readout that maximizes $\alpha$ (top) does not lead to the smallest $\beta$. (**C**) The values of $\alpha$ and $\beta$ accurately predict the preparation index ($R^2 = 0.93$) for a range of high-dimensional inhibition-stabilized networks (ISNs) (maroon dots) with different connectivity strengths and characteristic timescales (Methods). The best fit (after z-scoring) is given by $f(\alpha, \beta) = (16.94 \pm 0.02)\alpha - (15.97 \pm 0.02)\beta$ (mean ± s.e.m. were evaluated by boostrapping). This confirms our hypothesis that optimal control relies more on preparation when $\alpha$ is large and $\beta$ is small. Note that $\alpha$ and $\beta$ alone only account for 34.8% and 30.4% of the variance in the preparation index, respectively (inset). Thus, $\alpha$ and $\beta$ provide largely complementary information about the networks' ability to use inputs, and can be combined into a very good predictor of the preparation index. Importantly, even though this fit was obtained *using ISNs only*, it still captures 69% of preparation index variance across networks from other families (blue dots; Methods).

## Control-theoretic properties predict the amount of preparation

Our investigation of preparation in a low-dimensional system allowed us to isolate the impact of core dynamical motifs, and highlighted how preparation depends on the geometry of the flow field, and its alignment to the readout. However, these intuitions remain somewhat qualitative, making them difficult to generalize to our high-dimensional ISN model.

To quantify the key criteria that appear important for preparation, we turned to tools from control theory. We reasoned that, for a network to be able to benefit from preparation and thus exhibit a large preparation index, there must be some advantage to using early inputs that do not immediately cause movement, relative to using later inputs that do. We hypothesized that this advantage could be broken down into two criteria. First, there must exist activity patterns that are momentarily output-null (i.e. do not immediately cause movement) yet seed output-potent dynamics that subsequently move the arm. The necessity of this criterion was obvious in the 2D nonnormal network, which did not display any preparation when its nullspace was aligned with its 'sink' mode. In the language of control theory, this criterion implies that the nullspace of the readout must be sufficiently 'observable' – we captured this in a scalar quantity $\alpha$ (Methods; *Kao and Hennequin, 2019*; *Skogestad, 2007*). Second, there must be a sizeable cost to performing the movement in an entirely input-driven manner without relying on preparation. In other words, the network should be hard to steer along the readout direction, i.e., the readout must be of limited 'controllability' – we captured this in another scalar quantity $\beta$ (Methods).

We illustrate the meaning of these two metrics in *Figure 5A and B* for a 2D example network that combines nonnormality and oscillations. We show two extreme choices of readout direction (*Figure 5A*, dashed black): the one that maximizes $\alpha$ (top) and the one that minimizes it (bottom). In the first case, the readout nullspace (dashed orange) is very observable, i.e., trajectories that begin in the nullspace evolve to produce large transients along the readout (solid orange and inset). In the second case, the opposite is true. For each case, we also assessed the controllability of the readout ($\beta$). The controllability of a direction corresponds to how much variance activity trajectories exhibit along that direction, when they are randomly and isotropically initialized (*Figure 5B*). In other words, a very controllable direction is one along which network trajectories have a natural tendency to evolve.

We then assessed how well $\alpha$ and $\beta$ could predict the preparation index of individual networks. In 2D networks, we found that a simple function that grows with $\alpha$ and decreases with $\beta$ could accurately predict preparation across thousands of networks (Appendix 1 - Section 3 'Additional results in the 2D system'). To assess whether these insights carried over to high-dimensional networks, we then generated a range of large ISNs with parametrically varied connectivity strengths and decay timescales (Methods). We then regressed the preparation index against the values of $\alpha$ and $\beta$ computed for each of these networks (as controllability and observability are only defined for linear networks, we set $\phi(x) = x$ for this investigation). We found that a simple linear mapping, prep. index $= k_0 + k_\alpha \alpha + k_\beta \beta$, with parameters fitted to one half of the ISNs, accurately predicted the preparation indices of the other half (*Figure 5C*; $R^2 = 0.93$, fivefold cross-validated). Interestingly, we observed that although $\alpha$ and $\beta$ (which are both functions of the network connectivity) were highly correlated across different networks, discarding either variable in our linear regression led to a significant drop in $R^2$ (*Figure 5C*, inset). Importantly, it was their difference that best predicted the preparation index ($k_\alpha > 0$ and $k_\beta < 0$), consistent with our hypothesis that the preparation index is a relative quantity which increases as the nullspace becomes more observable, but decreases as readout dimensions become more controllable.

We were able to confirm the generality of this predictive model by generating networks with other types of connectivity (oscillatory networks, and networks with unstructured random weights), which displayed dynamics very different from the ISNs (see *Appendix 1—figure 6*). Interestingly, despite the different distribution of $\alpha$ and $\beta$ parameters in those networks, we could still capture a large fraction of the variance in their preparation index ($R^2 = 0.69$) using the linear fit obtained from the ISNs alone.

This confirms that $\alpha$ and $\beta$ can capture information about the networks' dynamics in a universal manner.

Note that we do not make any claims about the specific functional form of the relationship between $\alpha$, $\beta$, and the preparation index. Rather, we claim that there should be a broad trend for the preparation index to increase with $\alpha$ and decrease with $\beta$, and we acknowledge that this relationship could in general be nonlinear. Indeed, in 2D networks, we found that the preparation index was in fact better predicted by the ratio of $\alpha$ over $\beta$ than by their difference (*Appendix 1—figure 5*).

Finally, as the above results highlight that the amount of preparation depends on the alignment between internal dynamics and readout, one may wonder how much our conclusions depend on our use of a random unstructured readout matrix. First, we note that the effect of the alignment on preparation index is greatly amplified in the low-dimensional networks (*Figure 4G*). In high-dimensional networks, the null space of a random readout matrix $C$ will have some overlap with the most observable directions of the dynamics, thus encouraging preparation. Second, we performed additional simulations where we meta-optimized the readout so as to minimize the average optimal cost per movement. The resulting system is more observable overall (as it allows the network to solve the task at a lower cost) but relies just as much on preparation (*Appendix 1—figure 7*).

## Modeling movement sequences

Having gained a better understanding of what features lead a network to prepare, we next set out to assess whether optimal control could also explain the neural preparatory processes underlying the generation of movement *sequences*. We revisited the experimental studies of *Zimnik and Churchland, 2021*, where monkeys were trained to perform two consecutive reaches. Each trial started with the display of both targets, followed by an explicitly enforced delay period before the onset of the first reach. A distinction was made between 'double' reaches in which a pause was enforced between reaches, and 'compound' reaches in which no pause was required. This study concluded that, rather than the whole movement sequence unrolling from a single preparatory period, each reach was instead successively seeded by its own preparatory activity.

Here, we asked whether such an independent, successive preparation strategy would arise as an optimal control solution, in the same way that single-reach preparation did. Importantly, we could not answer this question by directly examining network inputs as we did for single reaches. Indeed, any network input observed before the second reach could be contributing either to the end of the first movement, or to the preparation of the next. In fact, the issue of teasing apart preparatory vs. movement-related activity patterns also arose in the analysis of the monkey data. To address this, *Zimnik and Churchland, 2021*, exploited the fact that monkey M1 activity just before and during single reaches is segregated into two distinct subspaces. Thus, momentary activity patterns (during either single or double reaches) can be unambiguously labeled as preparatory or movement-related depending on which of the two subspaces they occupied. We performed a similar analysis (Methods) and verified that preparatory and movement activity patterns in the model were also well segregated in their respective subspaces in the single-reach task (*Figure 6A and B*). We then assessed the occupancy of the preparatory subspace during double reaching in the model, and took this measure as a signature of preparation.

To model optimal control of a double reach, we modified our cost functional to account for the presence of two consecutive targets (see Methods). We considered the same set of eight targets as in our single-reach task, and modeled all possible combinations of two targets (one example shown in *Figure 6*). We set the hyper-parameters of the cost function such that both targets could be reached by the resulting optimal controller, in a way that matched important qualitative aspects of the monkeys' behavior (in particular, such that both reaches were performed at similar velocities, with the second reach lasting slightly longer on average; *Figure 6B and C*, top).

We projected the network activity onto preparatory and movement subspaces identified using single and double reaches activity (Methods). For double reaches with a long (600 ms) pause, the preparatory subspace was transiently occupied twice, with the two peaks occurring just before the onset of each movement in the sequence (*Figure 6B*, bottom).

Notably, the occupancy during the 'compound' reach (without pause; *Figure 6C*) also started rising prior to the first movement before decaying very slightly and peaking again before the second reach, indicating two independent preparatory events. This is somewhat surprising, given that a movement sequence can also be viewed as a single 'compound' movement, for which we have shown previously a unique preparatory phase is sufficient (*Figure 2*). In our model, this behavior can be understood to arise from the requirement that the hand stop briefly at the first target. To produce the second reach, the hand needs to accelerate again, which requires transient re-growth of activity in the network. Given that the network's dynamical repertoire exhibits limited timescales, this is most easily achieved by reinjecting inputs into the system.

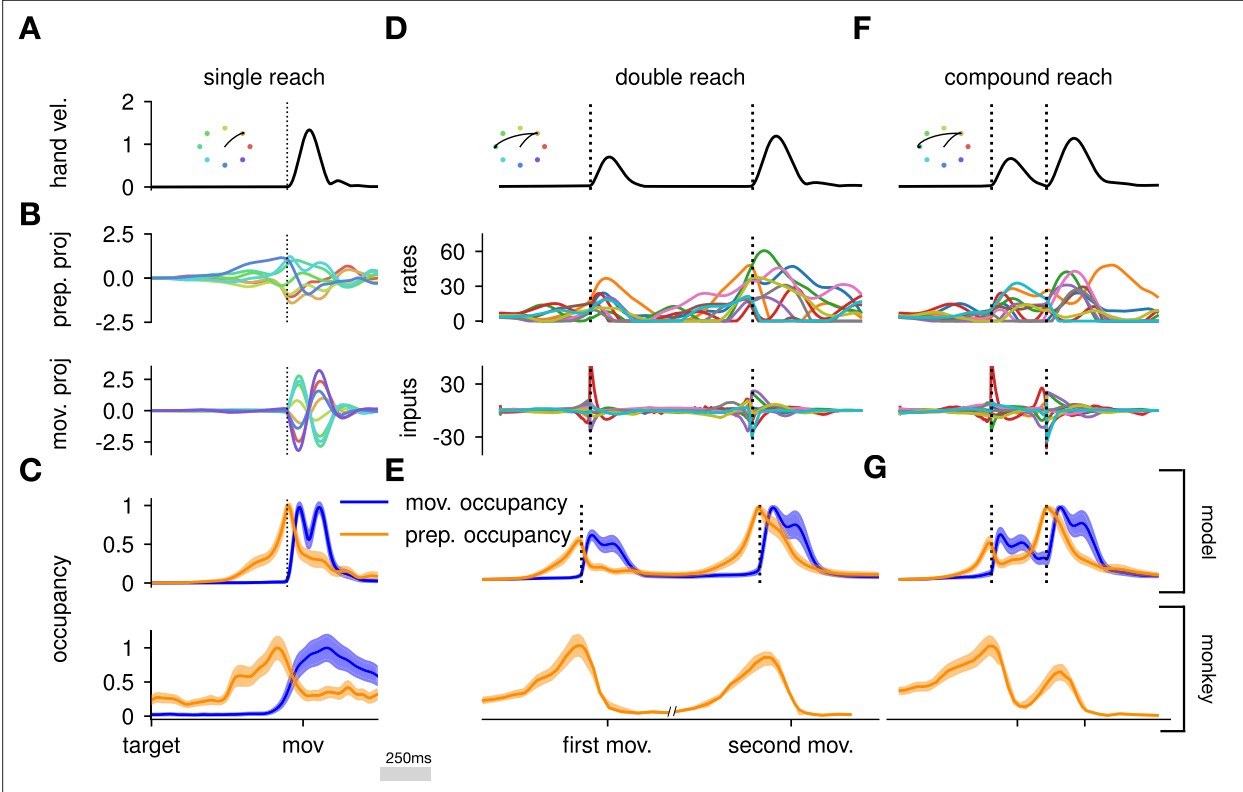

**Figure 6.** The model executes a sequence of two reaches using an independent strategy. (**A**) Hand velocity during one of the reaches, with the corresponding hand trajectory shown in the inset. (**B–C**) We identified two six-dimensional orthogonal subspaces, capturing 79% and 85% of total activity variance during single-reach preparation and movement respectively. (**B**) First principal component of the model activity for the eight different reaches projected into the subspaces identified using preparatory (top) and movement-epoch (bottom) activity. (**C**) Occupancy (total variance captured across movements) of the orthogonalized preparatory and movement subspaces, in the model (top) and in monkey motor cortical activity (bottom; reproduced from *Lara et al., 2018*, for monkey Ax). We report mean ± s.e.m., where the error is computed by bootstrapping from the neural population as in *Lara et al., 2018*. We normalize each curve separately to have a maximum mean value of 1. To align the model and monkey temporally, we re-defined the model's 'movement onset' time to be 120 ms after the model's hand velocity crossed a threshold – this accounts for cortico-spinal delays and muscle inertia in the monkey. Consistent with *Lara et al., 2018*'s monkey primary motor cortex (M1) recordings, preparatory subspace occupancy in the model peaks shortly before movement onset, rapidly dropping thereafter to give way to pronounced occupancy of the movement subspace. Conversely, there is little movement subspace occupancy during preparation. (**D**) Behavioral (top) and neural (middle) correlates of the delayed reach for one example of a double reach with an enforced pause of 0.6 s. The optimal strategy relies on preparatory inputs preceding each movement. (**E**) Same as (**C**), for double reaches. The onsets of the monkey's two reaches are separately aligned to the model's using the same convention as in (**C**). The preparatory subspace displays two clear peaks of occupancy. This double occupancy peak is also observed in monkey neural activity (bottom; reproduced from *Zimnik and Churchland, 2021*, with the first occupancy peak aligned to that of the model). (**F**) Same as (**D**), for compound reaches with no enforced pause in-between. Even though the sequence could be viewed as a single long movement, the control strategy relies on two periods of preparation. Here, inputs before the second reach are used to reinject energy into the system after slowing down at the end of the first reach. (**G**) Even though no explicit delay period is enforced in-between reaches during the compound movement, the preparatory occupancy rises twice, before the first reach and once again before the second reach. This is similar to observations in neural data (bottom; reproduced from *Zimnik and Churchland, 2021*).

In summary, our results suggest that the 'independent' preparation strategy observed in monkeys is consistent with the optimal control of a two-reach sequence. While *Zimnik and Churchland, 2021*, showed that RNNs trained on this task used this 'independent' strategy, this was by design as the network was only cued for the second reach after the first one had started. In addition to replicating this proof of concept that it is possible to prepare while moving, our model also shows how and why independent preparation might arise as an optimal control solution.

## Discussion

In this work, we proposed a model for the dynamics of motor cortex during a delayed-reaching task in non-human primates. Unlike previous work, we treated M1 as an input-driven nonlinear dynamical system, with generic connectivity not specifically optimized for the task, but with external inputs assumed to be optimal for each reach.

Motivated by a large body of evidence suggesting that preparation is useful before delayed reaches (*Churchland et al., 2010*; *Lara et al., 2018*; *Afshar et al., 2011*; *Shenoy et al., 2013*), but also evidence for thalamic inputs being necessary for accurate movement execution (*Sauerbrei et al., 2020*), we used this model to investigate whether and why neural circuits might rely on motor preparation during delayed-reaching tasks. Interestingly, preparation arose as an optimal control strategy in our model, with the optimal solution to the task relying strongly on inputs prior to movement onset. Moreover, the benefits of preparation were dependent on the network connectivity, with preparation being more prevalent in networks whose rich internal dynamics can be advantageously seeded by early external inputs. We were able to quantify this intuition with a predictive model relating the dynamical response properties of a network to the amount of preparation it exhibits when controlled optimally.

Finally, we found that prominent features of the monkeys' neural activity during sequential reaches arose naturally from optimal control assumptions. Specifically, optimally controlled networks relied on two phases of preparation when executing sequences of two reaches, corroborating recent experimental observations in monkey M1 (*Zimnik and Churchland, 2021*). Together, our results provide a normative explanation for the emergence of preparatory activity in both single and sequential reaching movements.

In recent years, task-optimized RNNs have become a very popular tool to model neural circuit dynamics. Classically, those models incorporate only those inputs that directly reflect task-related stimuli (e.g. motor target, go cue, etc.). This requires assumptions about the form of the inputs, such as modeling them as simple step functions active during specific task epochs. However, as local neural circuits are part of a wider network of brain areas, a large portion of their inputs come from other brain areas at intermediate stages of the computation and may therefore not be directly tied to task stimuli. Thus, it is not always obvious what assumptions can reasonably be made about the inputs that drive the circuit's dynamics.

Our optimization framework, which does not require us to make any specific assumptions about when and how inputs enter the network (although it does allow to incorporate prior information in the form of constraints), allows to bypass this problem and to implicitly model unobserved inputs from other areas. Importantly, our framework allows to ask questions – such as 'why prepare' – that are difficult to phrase in standard input-driven RNN models. We note, however, that in the investigation we have presented here, the lack of imposed structure for the inputs also implied that the model could not make use of mechanisms known to contribute certain aspects of preparatory neural activity. For example, our model did not exhibit the usual visually driven response to the target input, nor did it have to use the delay epoch to keep such a transient sensory input in memory (*Guo et al., 2014*; *Li et al., 2015*).

The main premise of our approach is that one can somehow delineate the dynamical system which M1 implements, and attribute any activity patterns that it cannot autonomously generate to external inputs. Just where the anatomical boundary of 'M1 dynamics' lie – and therefore where 'external inputs' originate – is unclear, and our results must be interpreted with this limitation in mind. Operationally, previous works in reaching monkeys have shown that M1 data can be mathematically well described by a dynamical system that appears largely autonomous during movement. These works have emphasized that those abstract dynamics, while inferred from M1 data alone, may not be anatomically confined to M1 itself. Instead, they may involve interactions between multiple brain areas, and even possibly parts of the body through delayed sensory feedback. Here, we too tend to think of our M1 models in this way, and therefore attribute external input to brain areas that are one step removed from this potentially broad motor-generating network. Nevertheless, a more detailed multi-area model of the motor-generating circuitry including, e.g., spinal networks (*Prut and Fetz, 1999*) could enable more detailed comparisons to multi-region neural data. In a similar vein, our model makes no distinction between external inputs that drive movement-specific planning computations, and other types of movement-unspecific inputs that might drive the transition from planning

to execution (e.g. 'trigger' inputs, *Kaufman et al., 2016*). Incorporating such distinctions (e.g. by temporally modulating the cost in individual input channels depending on specific task events, or by having separate channels for movement-unspecific inputs) might allow to ask more targeted questions about the role and provenance of external inputs.

A major limitation of our study is the specific choice of a quadratic penalty on the external input in our control objective. While there are possible justifications for such a cost (e.g. regularization of the dynamics to promote robustness of the control solution), its use here is mainly motivated by mathematical tractability. Other costs might be conceivably more relevant and might affect our results. For example, studies of motor cortex have long thought of its dynamics as converting relatively simple inputs reflecting high-level, temporally stable plans, into detailed, temporally varying motor commands. Thus, a potentially relevant form of a penalty for external inputs would be their temporal complexity. Such a penalty would have the advantage of encouraging a clearer separation between the inputs and the RNN activations; indeed, in our current model, we find that the optimal controls themselves have a temporal structure, part of which could be generated by a dynamical system and thus potentially absorbed into our 'M1 dynamics'. To address this, we note that our optimization framework can be adjusted to penalize the magnitude of the temporal *derivative* of the external input $\|\dot{\boldsymbol{u}}\|^2$, instead of $\|\boldsymbol{u}\|^2$. We experimented with this extension and found qualitatively different optimal inputs and M1 firing rates, which evolved more slowly and plateaued for sufficiently long preparation (*Appendix 1—figure 8A-D*) – this is in fact more consistent with monkey M1 data (e.g. *Elsayed et al., 2016*). Despite these qualitative difference in the specific form of preparation, our main conclusion stands that input-driven preparation continues to arise as an optimal solution (*Appendix 1—figure 8E-F*).

Another important assumption we have made is that the optimal controller is aware of the duration of the delay period. While this made solving for the optimal control inputs easier, it made our task more akin to a self-initiated reach (*Lara et al., 2018*) than to a typical delayed reach with unpredictable, stochastic delay durations. Future work could revisit this assumption. As a first step toward this, we now briefly outline pilot experiments in this direction. We used an exponential distribution of delays (with mean 300 ms) and devised two modified versions of our model that dealt with the resulting uncertainty in two different ways. In the first strategy, at any time during preparation, the model would estimate the most probable time-to-go-cue given that it hadn't arrived yet (in this case, this is always 300 ms in the future) and would plan an optimal sequence of inputs accordingly. In the second strategy, the network would prudently assume the earliest possible go cue (i.e. the next time step) and plan accordingly. In both cases, only the first input in the optimal input sequence would be used at each step, and complete replanning would follow in the next step, as the model re-assesses the situation given new information (i.e. whether the actual go cue arrived or not; this is a form of 'model predictive control', *Rawlings et al., 2017*). Preparatory inputs arose in both settings, but we found that only the latter strategy led to activity patterns that plateaued early during preparation (see *Appendix 1—figure 9*).

Throughout the main text, we have referred to $\Delta_{\mathrm{prep}}$ as the task-enforced delay period. However, a more accurate description may be that it corresponds to a delay period determined by an internally set go signal, which can lag behind the external go cue. While we would not expect a large difference between those two signals, the way in which we define $\Delta_{\mathrm{prep}}$ becomes important as it approaches 0 ms (limit of a quasi-automatic reach; *Lara et al., 2018*). Indeed, in this limit, our model exhibits almost no activity in the preparatory subspace (as defined in *Figure 6* – see further analyses in *Appendix 1—figure 10*). In contrast, monkey M1 activity was found to transiently occupy the preparatory subspace even in this case (*Lara et al., 2018*). Evidence for a delay between the earliest possible response to sensory cues and the trigger of movement was also observed in *Kaufman et al., 2016*, as well as in human behavioral studies (*Haith et al., 2016*). Thus, one may wish to explicitly incorporate this additional delay in the model in order to make it more realistic. Note however that *Haith et al., 2016*, showed that this internal delay could be shortened without affecting movement accuracy, suggesting that part of the processing that empirically occurs in-between the internal and external go cues may not be necessary, but rather reflect a decoupling between the end of preparation and the trigger of movement. This may be important to consider when attempting to compare the model to, e.g., reaction times from behavioral experiments.

Dynamical systems have a longstanding history as models of neural populations (*Dayan and Abbott, 2001*). However, understanding how neural circuits can perform various computations remains a challenging question.

Recently, there has been increased interest in trying to understand the role of inputs in shaping cortical dynamics. This question has been approached both from a data-driven perspective (*Malonis et al., 2021*; *Soldado-Magraner et al., 2023*) and in modeling work with, e.g., *Driscoll et al., 2022*, showing how a single network can perform different tasks by reorganizing its dynamics under the effect of external inputs and *Dubreuil et al., 2021*, relating network structure to the ability to process contextual inputs. To better understand how our motor system can generate flexible behaviors (*Logiaco et al., 2021*; *Stroud et al., 2018*), and to characterize learning on short timescales (*Sohn et al., 2021*; *Heald et al., 2023*), it is important to study how network dynamics can be modulated by external signals that allow rapid adaptation to new contexts without requiring extensive modifications of the network's connectivity. The optimal control approach we proposed here offers a way to systematically perform such evaluations, in a variety of tasks and under different assumptions regarding how inputs are allowed to impact the dynamics of the local circuit of interest. While our model's predictions will depend on, e.g., the choice of connectivity or the design of the cost function, an exciting direction for future work will be to obtain those parameters in a data-driven manner, for instance using recently developed methods to infer dynamics from data (*Pandarinath et al., 2018*; *Schimel et al., 2022*), and advances in inverse reinforcement learning and differentiable control (*Amos et al., 2018*) to infer the cost function that behavior optimizes. These could additionally be combined with more biomechanically realistic effectors, such as the differentiable arm models from *Codol et al., 2023*.

## Methods
### Experimental model and subject details

In *Figure 1*, we showed data from two primate datasets that were made available to us by Mark Churchland, Matthew Kaufman, and Krishna Shenoy. Details of animal care, surgery, electrophysiological recordings, and behavioral task have been reported previously in *Churchland et al., 2012*; *Kaufman et al., 2014* (see in particular the details associated with the J and N 'array' datasets). The subjects of this study, J and N, were two adult male macaque monkeys (*Macaca mulatta*). The animal protocols were approved by the Stanford University Institutional Animal Care and Use Committee. Both monkeys were trained to perform a delayed-reaching task on a fronto-parallel screen. At the beginning of each trial, they fixated on the center of the screen for some time, after which a target appeared on the screen. After a variable delay period (0–1000 ms), a go cue appeared instructing the monkeys to reach toward the target. Recordings were made in the PMd cortex and in the M1 using a pair of implanted 96-electrode arrays. In *Figure 6*, we also reproduced data from *Lara et al., 2018*, and *Zimnik and Churchland, 2021*. Details of animal care, surgery, electrophysiological recordings, and behavioral task for those data can be found in the Methods section of the respective papers.

### Arm model

To simulate reaching movements, we used the planar two-link arm model described in *Li and Todorov, 2004*. The two links have lengths $L_1$ and $L_2$, masses $M_1$ and $M_2$, and moments of inertia $I_1$ and $I_2$, respectively. The lower arm's center of mass is located a distance $D_2$ from the elbow. By considering the geometry of the upper and lower limb, the position of the hand and elbow can be written as vectors $\mathbf{y}_h(t)$ and $\mathbf{y}_e$ given by

$$\mathbf{y}_h = \begin{pmatrix} L_1 \cos\theta_1 + L_2 \cos(\theta_1 + \theta_2) \\ L_1 \sin\theta_1 + L_2 \sin(\theta_1 + \theta_2) \end{pmatrix} \text{ and}$$

$$y_e = \begin{pmatrix} L_1 \cos\theta_1 \\ L_1 \sin\theta_1 \end{pmatrix}.$$

(6)

The joint angles $\boldsymbol{\theta} = (\theta_1; \theta_2)^T$ evolve dynamically according to the differential equation

$$\boldsymbol{m}(t) = \mathcal{M}(\boldsymbol{\theta})\ddot{\boldsymbol{\theta}} + \mathcal{X}(\boldsymbol{\theta}, \dot{\boldsymbol{\theta}}) + \mathcal{B}\dot{\boldsymbol{\theta}},$$

(7)

where $\boldsymbol{m}(t)$ is the momentary torque vector, $\mathcal{M}$ is the matrix of inertia, $\mathcal{X}$ accounts for the centripetal and Coriolis forces, and $\mathcal{B}$ is a damping matrix representing joint friction. These parameters are given by

$$\mathcal{M}(\boldsymbol{\theta}) = \begin{pmatrix} a_1 + 2a_2 \cos\theta_2 & a_3 + a_2 \cos\theta_2 \\ a_3 + a_2 \cos\theta_2 & a_3 \end{pmatrix} \tag{8}$$

$$\mathcal{X}(\boldsymbol{\theta}, \dot{\boldsymbol{\theta}}) = a_2 \sin\theta_2 \begin{pmatrix} -\dot{\theta}_2(2\dot{\theta}_1 + \dot{\theta}_2) \\ \dot{\theta}_1^2 \end{pmatrix} \tag{9}$$

$$\mathcal{B} = \begin{pmatrix} 0.05 & 0.025 \\ 0.025 & 0.05 \end{pmatrix} \tag{10}$$

with $a_1 = I_1 + I_2 + M_2 L_1^2$, $a_2 = M_2 L_1 D_2$, and $a_3 = I_2$.

## iLQR algorithm

Throughout this work, we used the iLQR algorithm (*Li and Todorov, 2004*) to find the locally optimal inputs that minimize our cost function. iLQR is a trajectory optimization algorithm that can handle nonlinear dynamics and non-quadratic costs. iLQR works in an iterative manner, by linearizing the dynamics and performing a quadratic approximation of the cost at each iteration, thus turning the control problem into a local linear quadratic problem whose unique solution is found using LQR (*Kalman, 1960*). The LQR solver uses a highly efficient dynamic programming approach that exploits the sequential structure of the problem. Our implementation of iLQR (*Schimel et al., 2021*) followed from *Li and Todorov, 2004*, with the difference that we performed regularization of the local curvature matrix as recommended by *Tassa, 2011*.

## Generation of the high-dimensional readouts and networks

### Generation of inhibitory-stabilized networks

Simulations in *Figures 1, 3, 5, and 6* were conducted using ISNs. Those were generated according to the procedure described in *Hennequin et al., 2014*, with minor adjustments. In brief, we initialized strongly connected chaotic networks with sparse and log-normally distributed excitatory weights, and stabilized them through progressive $\mathcal{H}_2$-optimal adjustments of the inhibitory weights until the spectral abscissa of the connectivity matrix fell below 0.8. This yielded strongly connected but stable networks with a strong degree of nonnormality. When considering a larger range of ISNs (*Figure 5*), we independently varied both the variance of the distribution of initial excitatory weights and the spectral abscissa below which we stopped optimizing the inhibitory weights.

### Generation of additional networks in *Figure 5*

To assess the generality of our findings in *Figure 5*, we additionally generated randomly connected networks by sampling each weight from a Gaussian distribution with $\sigma = R/\sqrt{N}$, where the spectral radius $R$ was varied between 0 and 0.99. We also sampled skew-symmetric networks by drawing a random network $\boldsymbol{S}$ as above, and setting $\boldsymbol{W} = (\boldsymbol{S} - \boldsymbol{S}^T)/2$. We varied the radius $R$ of the $\boldsymbol{S}$ matrices between 0 and 5. Moreover, we considered diagonally shifted skew-symmetric networks $\boldsymbol{W} = (\boldsymbol{S} - \boldsymbol{S}^T)/2 + \lambda\boldsymbol{I}$, where $\lambda$ denotes the real part of all the eigenvalues and was varied between 0 and 0.8.

The elements of the readout matrix $\boldsymbol{C}$ mapping neural activity onto torques were drawn from a normal distribution with zero mean and standard deviation $\sigma_C = 0.05/\sqrt{N}$. This was chosen to ensure that firing rates of standard deviation on the order of 30 Hz would be decoded into torques of standard deviation ~2 N/m, which is the natural variation required for the production of the reaches we considered.

## Details of Figure 4

To more easily dissect the phenomena leading to the presence or absence of preparation, we turned to 2D linear networks in *Figure 4*. We modeled nonnormal networks with a connectivity $W = \begin{bmatrix} 0 & 0 \\ w & 0 \end{bmatrix}$ and oscillatory networks with connectivity $W = \begin{bmatrix} 0 & -w \\ w & 0 \end{bmatrix}$. The activity of the two units evolved as

$$\tau \dot{x}(t) = -x(t) + Wx(t) + u(t) \tag{11}$$

and directly influenced the acceleration of a 1D output $y(t)$ according to

$$\ddot{y}(t) = C_i x(t) \tag{12}$$

where $C_i = \begin{bmatrix} \cos\theta_C & \sin\theta_C \end{bmatrix}$ was a row matrix reading the activity of the network along an angle $\theta_C$ from the horizontal (first unit). Our setup aimed to mirror the reaching task studied in this work. We thus optimized inputs to minimize the following cost function:

$$
\begin{aligned}
\mathcal{J}[u] \quad =: &\underbrace{\int_0^T \|y(t) - y^\star\|^2 \frac{t^2}{T^2} \frac{dt}{T}}_{\mathcal{J}_{\text{target}}} \\
&+ \alpha_{\text{null}} \underbrace{\int_{-\Delta_{\text{prep}}}^0 \left( \|y(t)\|^2 + \|\dot{y}(t)\|^2 + \|\ddot{y}(t)\|^2 \right) \frac{dt}{T}}_{\mathcal{J}_{\text{null}}} \\
&+ \alpha_{\text{effort}} \underbrace{\int_{-\Delta_{\text{prep}}}^T \|u(t)\|^2 \frac{dt}{2T}}_{\mathcal{J}_{\text{effort}}} .
\end{aligned}
\tag{13}
$$

where $y^\star = 20$ was the target position.

## Computing networks' controllability and observability to predict preparation in Figure 5

As part of our attempt to predict how much a network will prepare given its intrinsic properties only, we computed the prospective potency of the nullspace α, and the controllability of the readout β. For a stable linear dynamical system described by

$$\frac{dx}{dt} = Ax(t) + Bu(t) \tag{14}$$

$$y(t) = C_x(t) \tag{15}$$

the system's observability Gramian $Q$ can be computed as the unique positive-definite solution of the Lyapunov equation

$$A^T Q + QA + C^T C = 0. \tag{16}$$

The prospective potency of the nullspace $C^\perp$ is then defined as

$$\alpha \triangleq \frac{\text{Tr}(C^\perp Q C^{\perp^T})}{N-2}. \tag{17}$$

Note that this measure $\alpha$ is invariant to the specific choice of basis for the nullspace $C^\perp$. Similarly, to assess the controllability of the readout, we first computed the controllability Gramian of the system $P$, which is the solution of

$$AP + PA^T + BB^T = 0, \tag{18}$$

with $B = I$ in our system. We then defined the controllability of the readout as

$$\beta \triangleq \frac{\text{Tr}(CPC^T)}{2}. \tag{19}$$

## Details of Figure 6

### Cost function

We modeled sequences of reaches by modifying our cost functional to account for the presence of two targets, as

$$
\mathcal{J}[\boldsymbol{u}] = \underbrace{\int_0^{\Delta_{\text{move}}^{(1)}+\tau} \|\boldsymbol{\theta}(t) - \boldsymbol{\theta}_1^\star\|^2 \frac{t^2}{T^2} dt}_{\mathcal{J}_{\text{target}}^{(1)}}
$$
$$
+ \alpha_{\text{pause}} \underbrace{\int_{\Delta_{\text{move}}^{(1)}}^{\Delta_{\text{move}}^{(1)}+\tau} \|\dot{\boldsymbol{\theta}}(t)\|^2 dt}_{\mathcal{J}_{\text{pause}}} \tag{20}
$$
$$
+ \underbrace{\int_{\Delta_{\text{move}}^{(1)}+\tau}^{T} \|\boldsymbol{\theta}(t) - \boldsymbol{\theta}_2^\star\|^2 \frac{(t - \Delta_{\text{move}}^{(1)} - \tau)^2}{T^2} dt}_{\mathcal{J}_{\text{target}}^{(2)}}
$$
$$
+ \alpha_{\text{null}} \underbrace{\int_{-\Delta_{\text{prep}}}^{0} \|\boldsymbol{\theta}(t) - \boldsymbol{\theta}_0\|^2 + \|\dot{\boldsymbol{\theta}}(t)\|^2 + \|\boldsymbol{m}(t)\|^2 dt}_{\mathcal{J}_{\text{null}}} \tag{21}
$$
$$
+ \alpha_{\text{effort}} \underbrace{\int_{-\Delta_{\text{prep}}}^{T} \|\boldsymbol{u}(t)\|^2 dt}_{\mathcal{J}_{\text{effort}}}
$$

where $\tau$ describes how long the monkey's hands had to stay on the intermediate target before performing its second reach. We used $\tau = 600\,\text{ms}$ and $\alpha_{\text{pause}} = 100$ for the double reaches in which a pause was explicitly enforced during the experiment. For compound reaches, the experiment did not require monkeys to stop for any specific duration. However, to ensure that the hand stopped on the target in the model (as it does in experiments when monkeys touch the screen) rather than fly through it, we set $\tau = 6\,\text{ms}$ and $\alpha_{\text{pause}} = 100$ when modeling compound reaches.

### Preparatory subspace analysis

*Lara et al., 2018*, proposed an analysis to identify preparatory and movement-related subspaces. This analysis allows to assess when the neural activity enters those subspaces, independently of whether it is delay-period or post-go-cue activity.

The method identifies a set of preparatory dimensions and a set of movement dimensions, constrained to be orthogonal to one another, as in *Elsayed et al., 2016*. These are found in the following manner: the trial-averaged neural activity is split between preparatory and movement-related epochs, yielding two matrices of size $N \times MT$, where $N$ is the number of neurons, $T$ is the number of time bins, and $M$ is the number of reaches. One then optimizes the $W_{\text{prep}} \in \mathbb{R}^{N \times d_{\text{prep}}}$ and

$W_{\text{mov}} \in \mathbb{R}^{N \times d_{\text{mov}}}$ (where $d_{\text{prep}}$ and $d_{\text{mov}}$ are the predefined dimensions of the two subspaces) such that the subspaces respectively capture most variance in the preparatory and movement activities, while being orthogonal to one another. This is achieved by maximizing the following objective:

$$
\mathcal{C}(W_{\text{prep}}, W_{\text{mov}}) = \frac{1}{2}\left( \frac{\text{Tr}(W_{\text{prep}}^T C_{\text{prep}} W_{\text{prep}})}{Z_{\text{prep}}(d_{\text{prep}})} + \frac{\text{Tr}(W_{\text{mov}}^T C_{\text{mov}} W_{\text{mov}})}{Z_{\text{mov}}(d_{\text{mov}})} \right) \tag{22}
$$

where $C_{\text{prep/mov}}$ are the covariance matrices of the neural activity during the preparatory and movement epochs, respectively. The normalizing constant $Z_{\text{prep}}(d_{\text{prep}})$ denotes the maximum amount of variance in preparatory activity that can be captured by any subspace of dimension $d_{\text{prep}}$ (this is found via SVD), and similarly for $Z_{\text{mov}}(d_{\text{mov}})$. The objective is maximized under the constraints $W_{\text{prep}}^T W_{\text{mov}} = 0$, $W_{\text{prep}}^T W_{\text{prep}} = I$, and $W_{\text{mov}}^T W_{\text{mov}} = I$. We set subspace dimensions $d_{\text{prep}} = d_{\text{mov}} = 6$, although our results were robust to this choice.

The occupancy of the preparatory subspace was defined as

$$\text{occupancy}^{\text{prep}}(t) = \sum_{k=1}^{d_{\text{prep}}} \text{var}_\theta(x_k^{\text{prep}}(t, \theta))$$

and that of the movement subspace was defined as

$$\text{occupancy}^{\text{mov}}(t) = \sum_{k=1}^{d_{\text{mov}}} \text{var}_\theta(x_k^{\text{mov}}(t, \theta)).$$

For single reaches, we defined preparatory epoch responses as the activity in a 300 ms window before the end of the delay period, and movement-epoch responses as the activity in a 300 ms window starting 50 ms after the go cue. We normalized all neural activity traces using the same procedure as *Churchland et al., 2012*; *Elsayed et al., 2016*. For double reaches, we followed *Zimnik and Churchland, 2021*, and used neural activity traces from both single reaches and the first reach of double-reach sequences. Note that we did not include any activity from the second reaches in the sequence, or from compound reaches, when defining the subspaces.

## Parameter table

Parameters used for the various simulations.

| Symbol | Figure 1 | Figure 2 | Figure 3 | Figure 5 | Figure 4 | Figure 6 | Unit | Description |
|---|---|---|---|---|---|---|---|---|
| $L_1$ | 30 | | | | – | 30 | cm | Length of the upper arm in model |
| $L_2$ | 30 | | | | - | 30 | cm | Length of the forearm in model |
| $I_1$ | 0.025 | | | | – | 0.025 | kg/m$^{-2}$ | Inertia of upper arm |
| $I_2$ | 0.045 | | | | - | 0.045 | kg/m$^{-2}$ | Inertia of forearm |
| $M_1$ | 1.4 | | | | – | 1.4 | kg | Mass of upper arm |
| $M_2$ | 1.0 | | | | - | 1.0 | kg | Mass of forearm |
| $D_2$ | 16 | | | | – | 16 | cm | Elbow to lower arm center of mass distance |
| $r$ | 12 | | | | 20 | 12 | cm | Radius of the target reach |
| $\mu_{\mathbf{h}}$ | 20 | | | | 0 | – | mV | Mean baseline firing rate |
| $\sigma_{\mathbf{h}}$ | 5 | | | | 0 | - | mV | s.t.d of the baseline firing rate |
| $\alpha_{\text{effort}}$ | 5E-7 | | | | 1E-5 | 5E-7 | – | Coeff. of input cost |
| $\alpha_{\text{null}}$ | 1 | | | | 1 | 10 | - | Coeff. of cost of moving during the delay |
| $\alpha_{\text{pause}}$ | - | | | | | 100 | – | Coeff. of cost of moving between reaches |
| $\tau$ | 150 | | | | | | ms | Single-neuron time constant |
| $\Delta_{\text{move}}^{(1)}$ | – | | | | | 300 | ms | Duration of the first reach |
| $\Delta_{\text{prep}}$ | 500 | 300 | - | 300 | | 500 | ms | Delay period time |

*Continued on next page*

*Continued*

| Symbol | Figure 1 | Figure 2 | Figure 3 | Figure 5 | Figure 4 | Figure 6 | Unit | Description |
|---|---|---|---|---|---|---|---|---|
| $T$ | 1100 | 900 | – | 900 | | 2000—1406 | ms | Total movement duration |
| $N$ | 200 | | | | - | 200 | - | Number of neurons |
| $p_{con}$ | 0.2 | | | | – | 0.2 | – | Connection probability (E neurons) |
| $p_E$ | 80 | | | | - | 80 | - | Percentage of E neurons |
| $p_I$ | 20 | | | | – | 20 | – | Percentage of I neurons |

# Acknowledgements

We are grateful to Matthew T Kaufman and Mark M Churchland for sharing data for the monkey experiments. We thank Kristopher Jensen, David Liu, Javier Antorán, and Rory Byrne for helpful comments on the manuscript. MS was funded by an EPSRC DTP studentship, and part of this work was performed using resources provided by the Cambridge Tier-2 system operated by the University of Cambridge Research Computing Service (http://www.hpc.cam.ac.uk) funded by EPSRC Tier-2 capital grant EP/P020259/1. For the purpose of open access, the authors have applied a Creative Commons Attribution (CC BY) licence to any Author Accepted Manuscript version arising from this submission.

# Additional information

## Funding

| Funder | Grant reference number | Author |
|---|---|---|
| Engineering and Physical Sciences Research Council | RG94782 | Marine Schimel |

The funders had no role in study design, data collection and interpretation, or the decision to submit the work for publication.

## Author contributions

Marine Schimel, Conceptualization, Formal analysis, Investigation, Visualization, Writing - original draft, Writing - review and editing; Ta-Chu Kao, Conceptualization; Guillaume Hennequin, Conceptualization, Supervision, Visualization, Writing - original draft, Writing - review and editing

## Author ORCIDs

Marine Schimel ⓘ https://orcid.org/0000-0002-6937-011X

Reviewer #1 (Public Review): https://doi.org/10.7554/eLife.89131.4.sa1
Reviewer #2 (Public Review): https://doi.org/10.7554/eLife.89131.4.sa2
Reviewer #3 (Public Review): https://doi.org/10.7554/eLife.89131.4.sa3
Author response https://doi.org/10.7554/eLife.89131.4.sa4

# Additional files

## Supplementary files
• MDAR checklist

## Data availability

The current manuscript is a computational study, so no data have been generated for this manuscript. Modelling code and the code used to generate figures and analyses is available at https://github.com/marineschimel/why-prep-2 (copy archived at *Schimel, 2024*).

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

# Appendix 1

## A1.1 Choice of the hyperparameters of the model

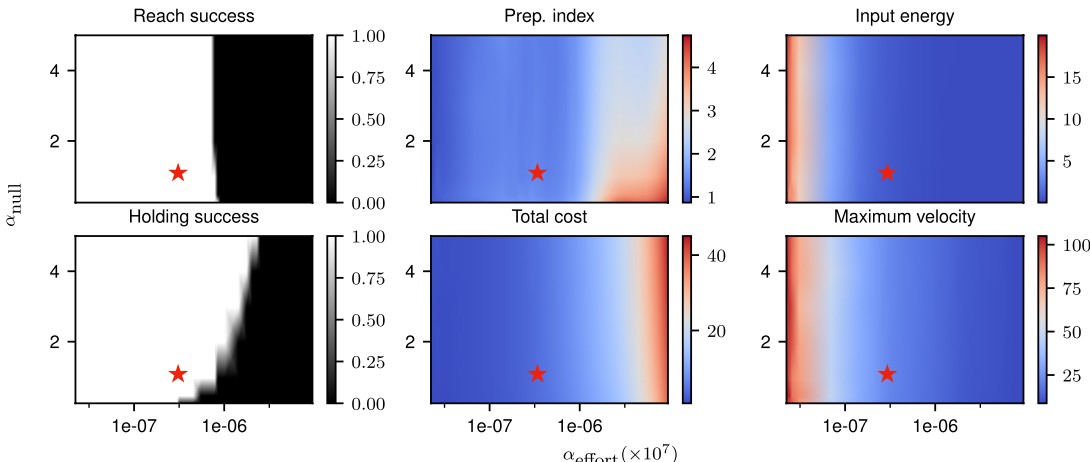

**Appendix 1—figure 1.** Correlates of the behavior and control strategy across a wide range of hyperparameters. The 'reach success'" and 'holding success'" are set to 1 if the success criterion (see text) is satisfied and 0 otherwise. The task is executed successfully over a wide range of hyperparameters. The red star denotes the set of hyperparameters used in the main text simulations. This configuration was chosen to lie in a region in which the task can be successfully solved, with the performance being robust to small changes in the hyperparameters.

Our cost function for the delayed single-reaching task was composed of three components. The relative weighings of the different terms in our cost, which are hyperparameters of the model, affect the way in which the task is solved. To ensure robustness of our results to hyperparameter changes, we scanned the space of $\alpha_{\text{null}}$ and $\alpha_{\text{effort}}$ (as the solution is invariant to scaling of the cost, only those relative weighings matter), and evaluated the solutions found across this hyperparameter space for a delayed reach of 300 ms.

Our evaluation was based on multiple criteria. We considered the target to have been successfully reached if the mean distance to the target in the last 200 ms of the movement was lower than 5 mm (for a reach radius of 12 cm). We considered that the requirement to stay still during the delay period was satisfied if the mean torques during preparation were smaller than 0.02 N/m. We computed the preparation index and total cost as described in **Equations 4 and 5**. We moreover computed the total input energy per neuron as $\frac{1}{N}\int_{-\Delta_{prep}}^{T}\|u\|^2 dt$, and the maximum velocity as $\max_t\sqrt{\dot{x}(t)^2+\dot{y}(t)^2}$. These various quantities are shown for a range of hyperparameters in **Appendix 1—figure 1**, with the choice of hyperparameters used throughout our simulations marked with a red star. This shows that the behavior of the model is consistent across a range of hyperparameter settings around the one we used.

In **Appendix 1—figure 2**, we illustrate the output of the model for several hyperparameter settings. One can notice that for very small values of $\alpha_{\text{effort}}$ the reach is successful, but executed with larger torques and velocity than is necessary – e.g., the red and yellow reaches are equally successful but the red one is much faster – which comes at the cost of larger inputs. We chose the set of hyperparameters for our simulations such as to lie in an intermediate regime in which the task is solved successfully, but without requiring more inputs than necessary.

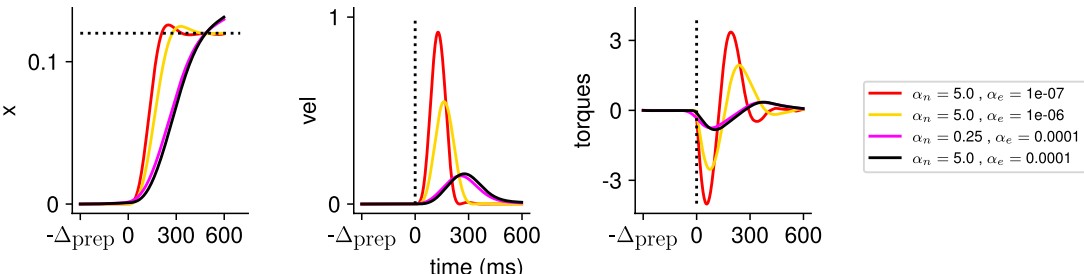

**Appendix 1—figure 2.** Illustration of the behavior for several hyperparameter settings. (Left) Hand position along the horizontal axis, with the dotted line denoting the position of the target. (Middle) Temporal profile of the hand velocity. (Right) Temporal profile of the torques driving the hand.

## A1.2 Investigation of the effect of the network decay timescale

*Appendix 1—figure 3* highlighted that preparatory inputs tend to consistently arise late during the delay period. We hypothesized that this may be a reflection of the intrinsic tendency of the network dynamics to decay, such that inputs given too early may be 'lost'. To test this, we changed the characteristic timescale of the dynamics *during preparation only*, leading to the following dynamics:

$$\tau_t \frac{dx(t)}{dt} = -x(t) + W\phi\left[x(t)\right] + h + \frac{\tau_t}{\tau_{\text{mov}}} u(t) \text{ where } \begin{cases} \tau_t = \tau_{\text{prep}} \text{if } t \leq 0 \\ \tau_t = \tau_{\text{mov}} \text{if } t \geq 0 \end{cases} \quad \text{(A1.1)}$$

with $\tau_{\text{mov}} = 150\,\text{ms}$. This allowed us to evaluate whether having dynamics decaying more slowly during preparation led to inputs starting earlier. Note that we also rescaled the inputs during preparation by $\frac{\tau_{\text{prep}}}{\tau_{\text{mov}}}$, to ensure that the effective cost of the inputs was not affected by the timescale change.

As shown in *Appendix 1—figure 3*, inputs started rising earlier when the network's decay timescale was longer. This was consistent with the hypothesis that the length of the window of preparation that the optimal controller uses depends on the network's intrinsic timescale.

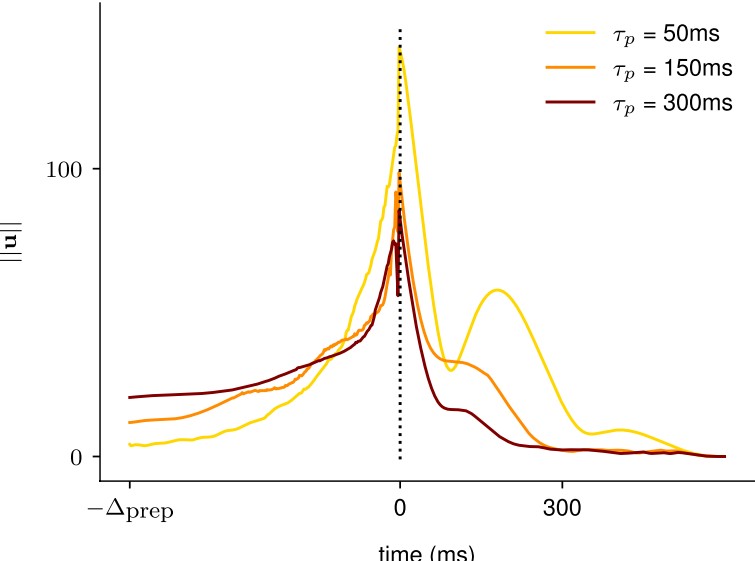

**Appendix 1—figure 3.** Illustration of the effect of the characteristic neuronal timescale on the temporal distribution of the inputs. We modified the characteristic neuronal timescale of the inhibition-stabilized network (ISN) during preparation only and assessed how that changed the temporal distribution of inputs for three different timescales ($\tau_{\text{prep}} = 50$ ms, $\tau_{\text{prep}} = 150$ ms, $\tau_{\text{prep}} = 300$ ms, top to bottom). As hypothesized, inputs start earlier during the preparation window when the decay timescale of the network was longer.

## A1.3 Additional results in the 2D system

Our visualization of the behavior of 2D networks in *Appendix 1—figure 4* allowed us to identify features of the dynamics that were well suited to predicting preparation. Below, we compute $\alpha$ and $\beta$ numerically and analytically in 2D oscillatory and nonnormal networks, to gain insights into how these quantities vary with the networks' dynamics. We then show how preparation can be predicted highly accurately across a large number of 2D systems, using only those quantities to summarize the network dynamics.

### A1.3.1 Controllability and observability computations

In *Appendix 1—figure 4*, we computed $\alpha$ and $\beta$ numerically, as a function of the connectivity strength and the choice of readout, for the nonnormal and the oscillatory motifs shown in *Appendix 1—figure 4*.

This highlights the very different behaviors of the two networks, which are to some extent also reflected in higher-dimensional models. In particular, we find a strong effect of the alignment between the readout and the network dynamics in nonnormal networks, while $\alpha$ and $\beta$ are independent of $\theta_C$ in oscillatoryq networks. Interestingly, we see that $\beta$ is constant across all oscillatory networks, while $\alpha$ increases with $w$.

As the reduced 2D model is more amenable to mathematical analysis than its high-dimensional counterpart, we can gain further insights into the origin of these differences by computing $\alpha(w, \theta_C)$ and $\beta(w, \theta_C)$ analytically.

Recall that the observability Gramian $Q$ of a linear input-driven dynamical system satisfies

$$A^T Q + QA + C^T C = 0 \tag{A1.2}$$

and the controllability Gramian satisfies

$$AP + PA^T + BB^T = 0, \tag{A1.3}$$

and that we defined $\alpha = \mathrm{Tr}(C^\perp Q C^{\perp T})$ and $\beta = \mathrm{Tr}(CPC^T)$, where $C^\perp$ denotes the nullspace of the readout matrix. Below, we compute these quantities for the 2D oscillatory and nonnormal networks, with $B = I$ and $C$ a unit-norm vector whose direction we parametrize via a quantity $\theta_C$. Note that we ignore the effect of $dt$ and $\tau$ in the mathematical analysis, as those quantities can straightforwardly be included in the final result via a rescaling of $w$ and $B$.

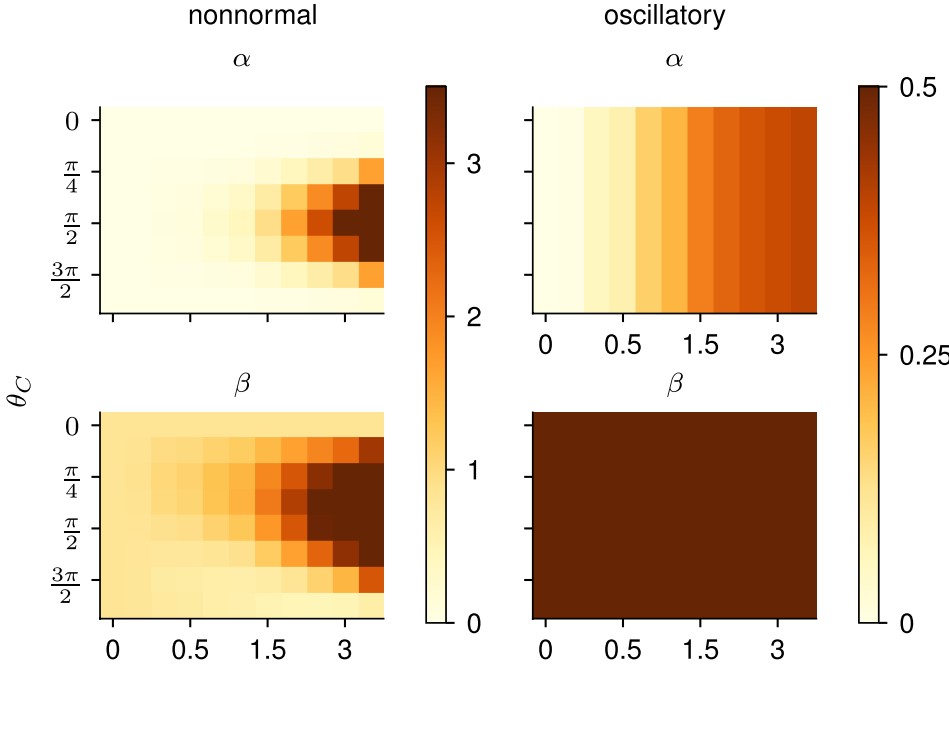

**Appendix 1—figure 4.** Illustration of α and β as a function of $\theta_C$ and $w$ in the 2D networks.

## Oscillatory network

In the case of $A = -I + S$, where $S$ is a skew-symmetric network (i.e. $S^T = -S$), **Equation A1.3** is solved by $P = I/2$ independently of the value of $S$. This explains why $\beta = \mathrm{Tr}(CPC^T) = \frac{1}{2}\|C\|^2$ is independent of both the connectivity strength $w$ and the orientation of the readout $\theta_C$ for skew-symmetric networks (see **Appendix 1—figure 4**, bottom right). Practically, this means that skew-symmetric networks are equally controllable in all directions: when driven by random inputs, these networks display isotropic activity of equal variance along all directions. Moreover, as $w$ controls the oscillation frequency of the network, but does not change the decay timescale of the eigenmodes, the amount of variance generated by a random stimulation is independent of $w$. Interestingly, we can see in **Appendix 1—figure 4** (top right) that $\alpha$ displays a different behavior, and increases with $w$. As highlighted above, skew-symmetric systems are rotationally symmetric. Without loss of generality, we can thus define our 1D vector to read out the first unit, i.e., C= $\begin{bmatrix} 1 & 0 \end{bmatrix}$.

The observability Gramian must satisfy

$$A^T Q + Q A + C^T C = 0 \implies \begin{bmatrix} -1 & w \\ -w & -1 \end{bmatrix} Q + Q \begin{bmatrix} -1 & -w \\ w & -1 \end{bmatrix} = \begin{bmatrix} -1 & 0 \\ 0 & 0 \end{bmatrix}. \tag{A1.4}$$

This can be found in closed form by solving the 2D system of equations, yielding

$$Q = \begin{bmatrix} \dfrac{1}{4} + \dfrac{1}{4(1+w^2)} & -\dfrac{w}{4(1+w^2)} \\ -\dfrac{w}{4(1+w^2)} & \dfrac{w^2}{4(1+w^2)} \end{bmatrix}. \tag{A1.5}$$

From there, we obtain $\alpha = \mathrm{Tr}(C^\perp Q C^{\perp^T}) = \frac{w^2}{4(1+w^2)}$. As found empirically, this quantity will initially increase before plateauing toward 1/4 as $w$ becomes large.

One might wonder why observability displays such a dependence on the oscillatory frequency of the network, even though the network is rotationally symmetric, and $w$ does not affect the decay timescale. As highlighted in **Equation A1.2**, controllability and observability Gramian would be identical for a skew-symmetric system if $C = I$. However, a feature of the systems we consider is

the existence of a nullspace, i.e., the fact that the readout $C$ only targets a subset of dimensions across the whole space (implying that $C^T C$ is a low-rank matrix). Intuitively, the reason why $\alpha$ increases with $w$ while $\beta$ is constant in skew-symmetric networks can be understood as follows: $\alpha$ is computing how much *readout activity* a vector initialized in the nullspace of $C$ will generate, while $\beta$ is computing the amount of energy that will be generated *across all directions* by a vector initialized in the readout space. Thus, assuming once again $C = \begin{bmatrix} 1 & 0 \end{bmatrix}$ and $C^\perp = \begin{bmatrix} 0 & 1 \end{bmatrix}$, the activity of vectors initialized along $C$ and $C^\perp$ respectively and evolving autonomously from there is given by $v_C(t) = \begin{bmatrix} e^{-t}\cos(wt) & e^{-t}\sin(wt) \end{bmatrix}$ and $v_{C^\perp}(t) = \begin{bmatrix} -e^{-t}\sin(wt) & e^{-t}\cos(wt). \end{bmatrix}$

From there, we can compute $\beta = \int_0^\infty \|v_C(t)\|^2 dt = \int_0^\infty e^{-2t} dt = \frac{1}{2}$. Thus, as found above, only the decay timescale of the envelope (fixed to 1 here) affects the value of $\beta$.

Importantly, $\alpha$ will instead have a dependence on $w$ arising from the fact that it depends on the size of the activity *projected into the readout*, as

$$\alpha = \int_0^\infty \|v_{C^\perp}(t)^T C\|^2 dt = \int_0^\infty e^{-2t}\sin^2(wt)dt \tag{A1.6}$$

$$= \frac{1}{2}\int_0^\infty e^{-2t}(1 - \cos(2wt))dt \tag{A1.7}$$

$$= \frac{1}{2}\int_0^\infty e^{-2t} - \Re e^{-2(1-iw)t} dt \tag{A1.8}$$

$$= \frac{1}{4} - \frac{1}{4(1+w^2)} \tag{A1.9}$$

$$= \frac{w^2}{4(1+w^2)}. \tag{A1.10}$$

The dependence of this quantity on $w$ can be understood by the fact that activity patterns initialized in the readout nullspace benefit from the existence of rotational dynamics, which allows them to be readout before the activity decays completely.

## Nonnormal network

In the nonnormal network, we have $A = -I + W = \begin{bmatrix} -1 & 0 \\ w & -1 \end{bmatrix}$. The nonnormal 2D system, unlike its oscillatory counterpart, does not have rotational symmetry. Thus, to remain general, we will consider $C(\theta_C) = \begin{bmatrix} \cos\theta_C & \sin\theta_C \end{bmatrix}$, and $C^\perp(\theta_C) = \begin{bmatrix} -\sin\theta_C & \cos\theta_C \end{bmatrix}$. Solving *Equation A1.3* for $B = I$ leads to an expression for the controllability Gramian of the nonnormal system as

$$P(w) = \begin{bmatrix} \dfrac{1}{2} & \dfrac{w}{4} \\ \dfrac{w}{4} & \dfrac{1}{2} + \dfrac{w^2}{4} \end{bmatrix}. \tag{A1.11}$$

Similarly, computation of the observability Gramian leads to

$$Q(w, \theta_C) = \begin{bmatrix} \dfrac{w^2\sin^2\theta_C}{4} + \dfrac{\cos\theta_C\sin\theta_C w}{2} + \dfrac{\cos^2\theta_C}{2} & \dfrac{w\sin^2\theta_C}{4} + \dfrac{\cos\theta_C\sin\theta_C}{2} \\ \dfrac{w\sin^2\theta_C}{4} + \dfrac{\cos\theta_C\sin\theta_C}{2} & \dfrac{\sin^2\theta_C}{2} \end{bmatrix}. \tag{A1.12}$$

We can then compute

$$\alpha(\theta_C, w) = C^{\perp T} Q C^\perp = \frac{w^2}{4}\sin^4\theta_C \tag{A1.13}$$

and

$$\beta(\theta_C, w) = \boldsymbol{C}^T \boldsymbol{P} \boldsymbol{C} = \frac{(w \sin \theta_C + \cos \theta_C)^2 - \cos^2 \theta_C + 2}{4}. \tag{A1.14}$$

This highlights the dependence of $\alpha$ and $\beta$ on $\theta_C$, which can also be seen in **Appendix 1—figure 4** (left). Interestingly, these expressions also make evident the supralinear scaling of $\alpha$ and $\beta$ with $w$ in nonnormal networks. Note however that we never investigate preparation in the very large $w$ regime, as the simulation of such networks with discretized dynamics is prone to numerical issues.

### A.13.2 Predicting preparation in 2D networks

To assess how well preparation could be predicted from the control-theoretic properties $\alpha$ and $\beta$ (c.f. main text) of 2D networks, we generated 20,000 networks with weight matrix

$$\boldsymbol{W}(\mathrm{a}, \omega, w_{\mathrm{ff}}) = \begin{bmatrix} a & \frac{1}{2}(w_{\mathrm{ff}} + \sqrt{w_{\mathrm{ff}}^2 + 4\omega^2}) \\ \frac{1}{2}(w_{\mathrm{ff}} - \sqrt{w_{\mathrm{ff}}^2 + 4\omega^2}) & a \end{bmatrix} \tag{A1.15}$$

where $a \sim \mathcal{U}(0, 0.8)$, $\omega \sim \mathcal{U}(0, 4)$, and $w_{\mathrm{ff}} \sim \mathcal{U}(0, 4)$. **Equation A1.5** implies that $W$ has a pair of complex-conjugate eigenvalues $a \pm i\omega$, and also embeds a feedforward coupling of strength $w_{\mathrm{ff}}$ from the second to the first dimension. For each network configuration, we computed the corresponding values of $\alpha$ and $\beta$. To confirm our intuition that the preparation index should increase with $\alpha$ and decrease with $\beta$, we first attempted to fit prep. index $= c_0 + c_1 \frac{\alpha}{\beta}$. Interestingly, we found that while this quantity was positively correlated with the preparation index across networks, a substantial fraction of variance remained unexplained (test $R^2 = 0.16$). Labeling the preparation index by the rotational frequency of the network highlighted that a substantial fraction of the variance across networks came from this timescale of oscillations (**Appendix 1—figure 5**, left). Indeed, a regression model of the prep. index $= c_0 + c_1 \omega \frac{\alpha}{\beta}$ captured 80% of the variance in preparation index, yielding an accurate fit across networks with only two free parameters (**Appendix 1—figure 5**, right).

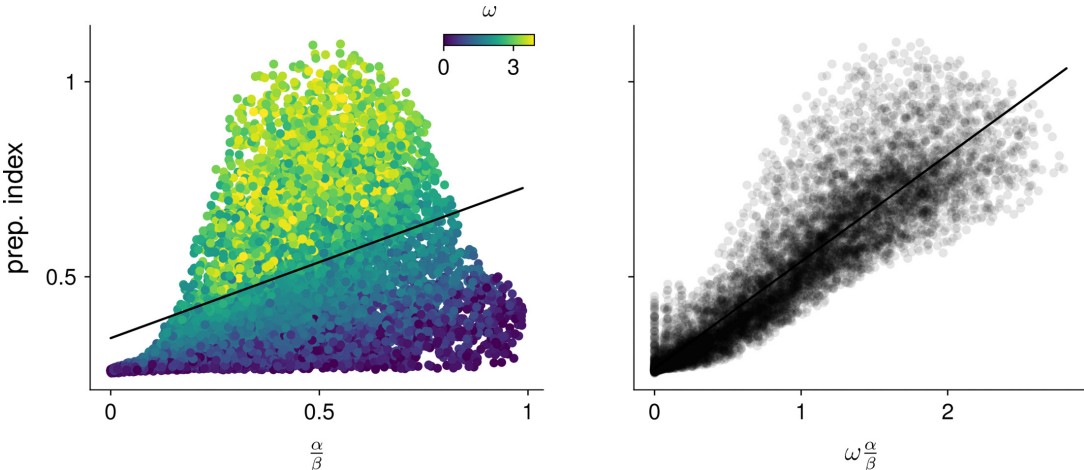

**Appendix 1—figure 5.** Predicting the preparation index from characteristic network quantities. We evaluated how well the preparation index could be predicted as a linear function of $\frac{\alpha}{\beta}$ (left). A substantial amount of residual variance appeared to arise from variability in the oscillation frequency $\omega$ (color). Accounting for this frequency by regressing the preparation index against $\omega \frac{\alpha}{\beta}$ gave a better fit (right).

We stress that the predictive power of these simple fits is remarkable given that the preparation index comes out of a complex process of optimization over control inputs. Thus, the control-theoretic quantities $\alpha$ and $\beta$ appear to appropriately summarize the benefits of preparation for individual networks.

The fact that the preparation index also grows with $\omega$ can be understood by considering the alignment between the activity trajectories which the network can autonomously generate and those that are required for solving the motor task. Indeed, a network that is intrinsically unable to generate outputs with the right oscillatory timescale would have to rely on movement-related inputs,

i.e., would have a low preparation index. As observed here, the network's characteristic frequency has a big impact in 2D networks, consistent with $\omega$ determining the *only* oscillatory pattern that the network can generate on its own. For high-dimensional networks, however, we did not have to incorporate such a measure of compatibility between task requirements and network dynamics (*Appendix 1—figure 5*). We speculate that this is due to averaging effects. Indeed, larger networks possess a wide range of intrinsic oscillatory timescales, and the readout matrix – which here was not aligned to the network's dynamics in any specific way – is expected to read out a little bit of all frequencies, including task-appropriate ones.

## A1.4 Comparison across networks

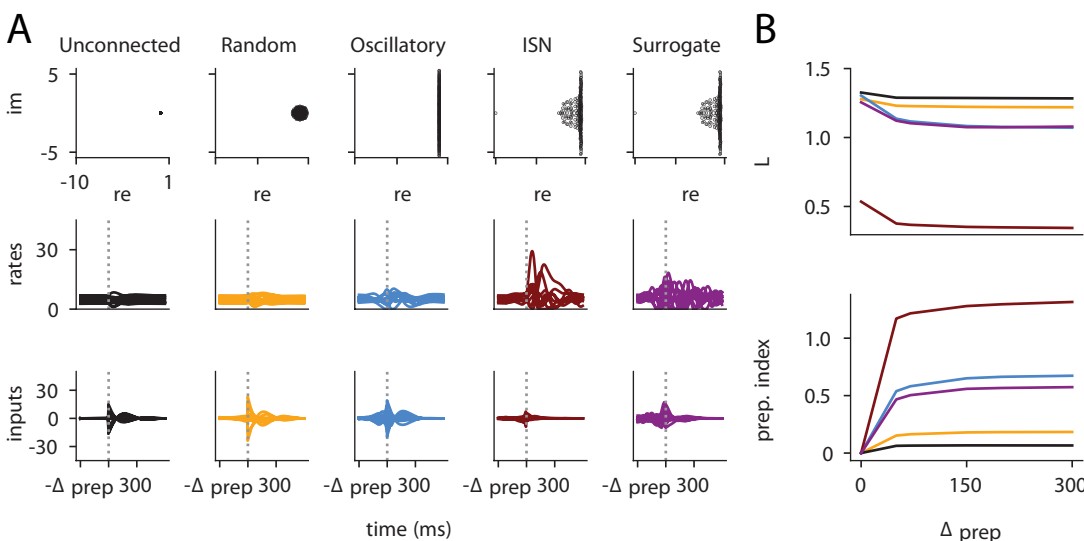

**Appendix 1—figure 6.** Preparation arises across a range of network architectures: neural correlates of the reach are shown for five different networks (**A**), alongside the loss and prep. index as a function of $\Delta_{\text{prep}}$. (**A**) Eigenvalue spectrum (top), internal network activations (middle), and inputs (bottom) for different network types. The unconnected network does not rely on preparatory inputs at all. The random network with weights draw from $\mathcal{N}(0, 0.95/\sqrt{N})$ uses very little delay-period inputs while the skew-symmetric network with $\mathcal{N}(0, 4/\sqrt{N})$ shows a substantial amount of inputs during the delay period. The inhibition-stabilized network can be seen to rely most on preparation, more so than the similarity transformed inhibition-stabilized network (ISN). (**B**) Loss (top) and preparation index (bottom) as a function of delay-period length for the different networks. The unconnected and random networks can be seen to benefit very little from longer preparation times. Indeed, even as $\Delta_{\text{prep}}$ increases, their amount of preparatory inputs remains very close to 0. On the other hand, the skew-symmetric network and the ISN use preparatory inputs (bottom), which allow them to have a lower loss for larger values of $\Delta_{\text{prep}}$. Interestingly, the surrogate ISN prepares considerably less than the full ISN.

Our main investigation was largely focused on behavior of inhibition-stabilized networks, which are believed to constitute good models of M1. We however found that the expression we derived to obtain a network's preparation index from its control-theoretic properties generalized across to other types of networks. Below, we detail the other network families we considered, and show how their dynamics qualitatively differ from the ISN, although their preparation can be predicted using the same quantities.

We modeled three additional classes of networks: randomly connected networks with either (i) unstructured or (ii) skew-symmetric connectivities, (iii) a surrogate network obtained by applying a similarity transformation to the ISN that preserved its eigenvalue spectrum but eliminated any 'nonnormality' (i.e., we found $\mathcal{T}$ such that $\tilde{A} = \mathcal{T}^{-1}A\mathcal{T}$, where $\tilde{A}$ was a diagonal matrix with the same eigenvalues as $A$). Note that we did not apply the transformation to the readout or input matrices, such that the transfer function of the system was changed by our transformation. This was voluntary, as we were interested in the effect that transforming the dynamics would have on the input-output response. These networks were chosen for the diversity of dynamical motifs they exhibit: combinations of rapidly and slowly decaying modes, oscillations, and transient dynamics.

Moreover, each of these network families could be sampled from in a straightforward manner, allowing to compute results across many instantiations of each network type. We again used random readout matrices not specifically adjusted to the dynamics of the network nor to the motor task.

To get an intuition for how different networks solve the task, we generated one network from each family and qualitatively compared their inputs and internal activations when performing the same delayed reach (*Appendix 1—figure 6A*). We first considered an unconnected network, i.e., a network whose recurrent weights were all 0. Unsurprisingly, this network had no use for a preparation phase. Indeed, there is no benefit to giving early inputs as the network is unable to amplify them. More surprisingly, a random network with a much stronger connectivity – as can be seen in its eigenvalue spectrum forming a small ball of radius close to 1 (*Appendix 1—figure 6A*, top) – also displayed very little preparation. The strong, visually apparent similarity between the inputs to the random and unconnected networks suggests that the optimal way of controlling the random network relies largely on ignoring its internal dynamics and solving the task almost entirely in an input-driven regime. The example skew-symmetric network, which had imaginary eigenvalues only (ranging between –5.5 and 5.5), displayed considerably more preparation, but still relied on strong inputs during the movement phase that resembled those of the unconnected and random networks. Finally, the ISN relied much more on preparation; the small inputs it receives are strongly amplified into large activity patterns owing to its strong, nonnormal recurrent connectivity. Interestingly however, the similarity transformed ISN lost much of that ability to amplify inputs, instead displaying dynamics resembling that of the skew-symmetric network. This highlights the effect of the ISN's nonnormal dynamics in shaping the network's activity and optimal inputs.

Next, we assessed more directly how beneficial preparation was for the different networks. We evaluated how the total loss and preparation index evolved as a function of the delay-period length (*Appendix 1—figure 6B*). As expected, the control of networks that relied on preparation (skew-symmetric and ISN) benefited more from longer delays. The ISN has markedly lower control cost and higher preparation index than other networks, reflecting the fact that even weak (thus energetically cheap) inputs were sufficient to produce internal activity and thus output torques of the desired magnitude (*Appendix 1—figure 6A*, right).

The above results give a sense of the range of possible dynamics that different types of networks display. Interestingly, despite these differences, we showed in *Appendix 1—figure 5* that the preparation index could be predicted with a simple formula across all networks.

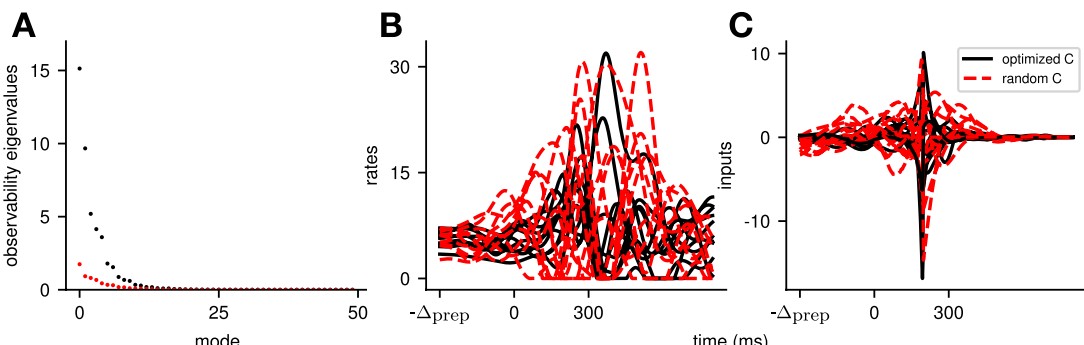

**Appendix 1—figure 7.** Illustration of the effect of optimizing the readout matrix such as to minimize the cost of the reaches, across all movements. To evaluate the effect that our choice of random readout directions has on our conclusions, we additionally compare to a model with the same dynamics, but where the readout was optimized such as to minimize the cost across movements (i.e. $\mathcal{L}(C) = \sum_{i \in \text{targets}} \mathcal{J}^{(i)}(C)$), under the constraint that its norm was fixed. In (**A**), we see that this leads to an increase in the observability of the system (compare the observability of the modes of the optimized system in black with those of the random readout in red). In (**B**) and (**C**), we see that this leads to an output of similar amplitude (**B**), but that is generated using smaller inputs (**C**). Importantly, we see that the system still relies on preparatory inputs. Thus, the exact choice of readout does not alter the network strategy, but can help the system perform the same movements in a more efficient manner.

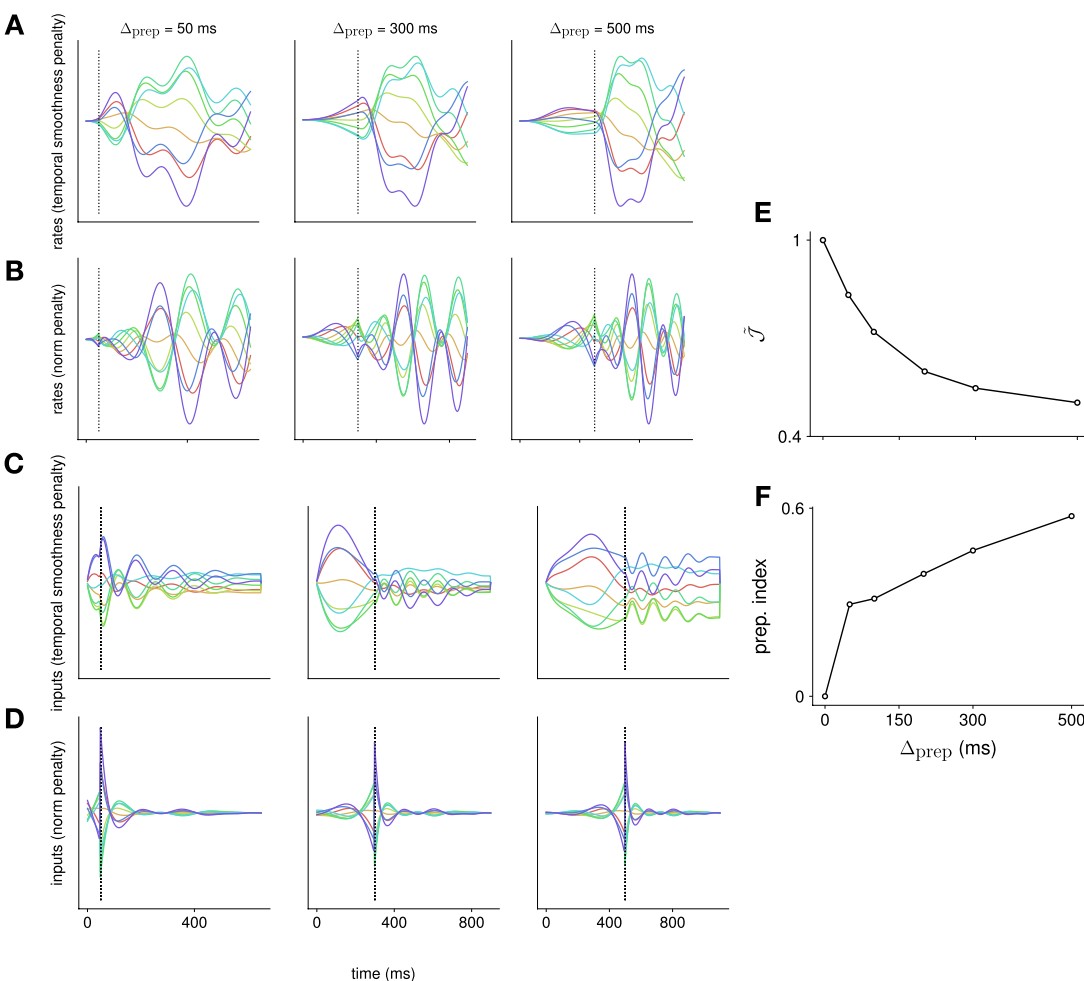

**Appendix 1—figure 8.** Comparison of the effect of penalizing temporal input smoothness vs. input norm. We compare the effect of using a cost over inputs that penalizes input norm, vs. using a cost that penalizes the 'temporal complexity' of the inputs – defined here as the temporal derivative of the inputs (i.e. $\|u(t+1) - u(t)\|^2$ in discrete time). This is achieved by augmenting the dynamical system to include an input integration stage, which then feeds into the original dynamical system; this way, the input to the augmented system – of which we continue to penalize the squared norm to enable the iterative linear quadratic regulator algorithm (iLQR) framework – is the derivative of the input to the original system. We perform this comparison in linear recurrent neural networks (RNNs), across a range of different preparation times. We show the activation of an example neuron in (**A**) and (**B**), and activity in an example input channel in (**C**) and (**D**). Each color denotes a different reach. We see that the rates vary more slowly when penalizing the temporal complexity (**A**) vs. the input norm (**B**), exhibiting a plateau for longer preparation times that is more similar to neural recordings. This is a reflection of the fact that the inputs themselves vary more slowly when the temporal complexity is penalized (compare **C** and **D**). As we do not penalize the input norm within our definition of temporal complexity, the optimal strategy is for the network to rely on steady inputs, which is different from the strategy used when the norm is penalized (compare **C** and **D**). We note that, under this different choice of input penalty, preparation nevertheless remains optimal, with the normalized loss (shown in **E**) decreasing as $\Delta_{\text{prep}}$ increases, and the preparation index (shown in **F**) increasing as $\Delta_{\text{prep}}$ increases.

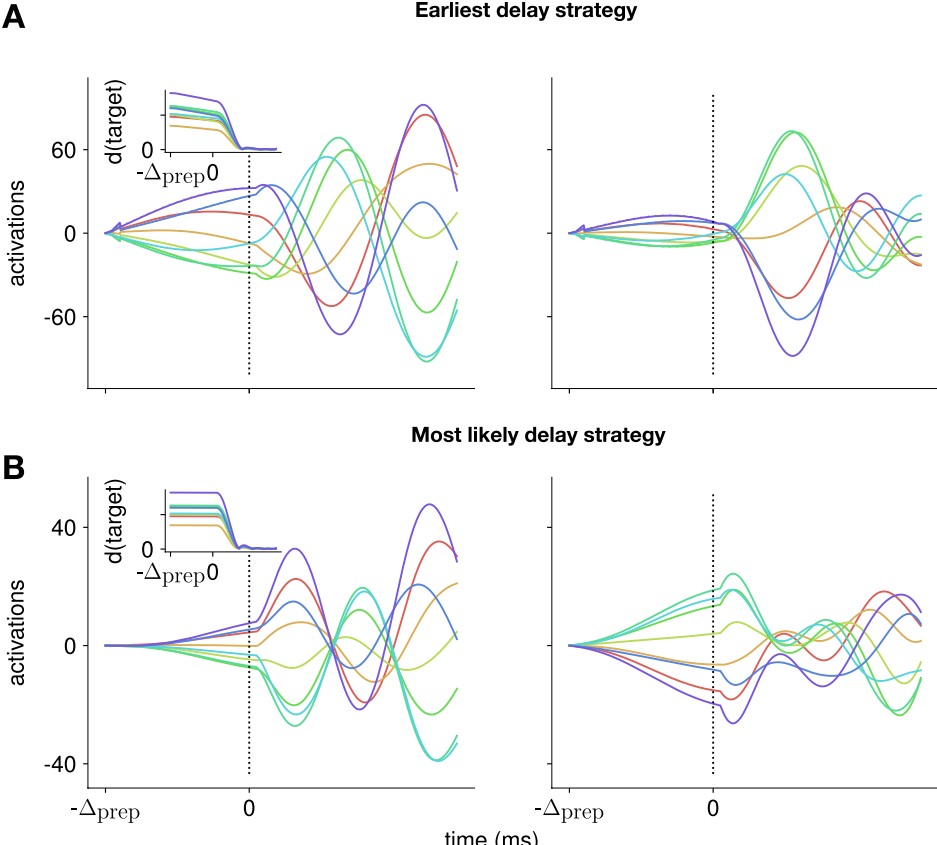

**Appendix 1—figure 9.** Control strategy adopted by the model when planning with uncertain delays. We investigate an extension of the model, that includes uncertainty about the arrival of the go cue, and involves replanning at regular intervals to update the model with new information (e.g. whether the go cue has arrived). In (**A**), we model this by assuming that the network is adopting a strategy whereby it plans to be ready to move *as early as possible* following the target onset. We optimize the inputs using this assumption, and replan every 20 ms to update the model with the available information (which here corresponds to the actual go cue only arriving 500 ms after target onset). We plot the activity of two example neurons (left and right panels, respectively), for each of the reaches (each color denotes a different reach). We can see that the neuronal activations start ramping up at the beginning of the task, and plateau before the actual target onset. In (**B**), we use a similar optimization strategy, but use a different 'mental model' for the network, whereby we assume that, until it sees the actual go cue, the model is always assuming that the delay period will be equal to the most likely a posteriori preparation time. Under the assumption of exponentially distributed delays with a mean of 150 ms, this corresponds to always replanning assuming a delay of 150 ms. We see that the network then adopts a different strategy, which does not include ramping/plateauing of the neural activity.

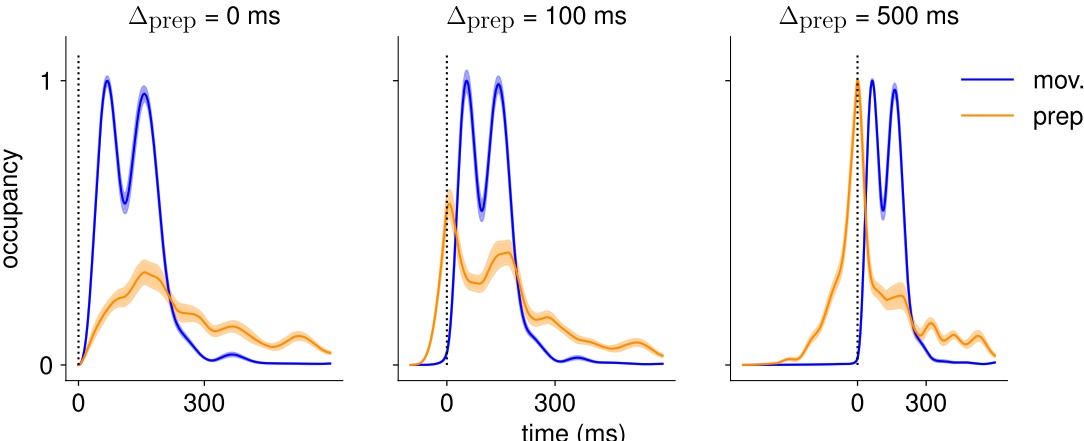

**Appendix 1—figure 10.** Comparison of the occupancy of the preparatory and movement subspaces across different delay periods. Occupancy (normalized by the maximum value across preparatory and movement occupancies) of the preparatory and movement subspaces identified using a delay period of 500 ms, for the activity generated using $\Delta_{\text{prep}} = 0$ ms (left), $\Delta_{\text{prep}} = 100$ ms (center), and $\Delta_{\text{prep}} = 500$ ms (right). We see that the network does not rely on preparatory activity when $\Delta_{\text{prep}} = 0$ ms.

