## [Editor Report · eLife assessment]

This **important** study provides a new perspective on why preparatory activity occurs before the onset of movement. The authors report that when there is a cost on the inputs, the optimal inputs should start before the desired network output for a wide variety of recurrent networks. The authors present **compelling** evidence by combining mathematically tractable analyses in linear networks and numerical simulation in nonlinear networks.

---

## [Referee Report · Reviewer #1 (Public Review)]

In this work, the authors investigate an important question - under what circumstances should a recurrent neural network optimised to produce motor control signals receive preparatory input before the initiation of a movement, even though it is possible to use inputs to drive activity just-in-time for movement?

This question is important because many studies across animal models have shown that preparatory activity is widespread in neural populations close to motor output (e.g. motor cortex / M1), but it isn't clear under what circumstances this preparation is advantageous for performance, especially since preparation could cause unwanted motor output during a delay.

They show that networks optimised under reasonable constraints (speed, accuracy, lack of pre-movement) will use the input to seed the state of the network before movement and that these inputs reduce the need for ongoing input during the movement. By examining many different parameters in simplified models they identify a strong connection between the structure of the network and the amount of preparation that is optimal for control - namely, that preparation has the most value when nullspaces are highly observable relative to the readout dimension and when the controllability of readout dimensions is low. They conclude by showing that their model predictions are consistent with the observation in monkey motor cortex that even when a sequence of two movements is known in advance, preparatory activity only arises shortly before movement initiation.

Overall, this study provides valuable theoretical insight into the role of preparation in neural populations that generate motor output, and by treating input to motor cortex as a signal that is optimised directly this work is able to sidestep many of the problematic questions relating to estimating the potential inputs to motor cortex.

---

## [Referee Report · Reviewer #2 (Public Review)]

This work clarifies neural mechanisms that can lead to a phenomenology consistent with motor preparation in its broader sense. In this context, motor preparation refers to activity that occurs before the corresponding movement. Another property often associated with preparatory activity is a correlation with global movement characteristics such as reach speed (Churchland et al., Neuron 2006), reach angle (Sun et al., Nature 2022), or grasp type (Meirhaeghe et al., Cell Reports 2023). Such activity has notably been observed in premotor and primary motor cortices, and it has been hypothesized to serve as an input to a motor execution circuit. The timing and mechanisms by which such 'preparatory' inputs are made available to motor execution circuits remain however unclear in general, especially in light of the presence of a 'trigger-like' signal that appears to relate to the transition from preparatory dynamics to execution activity (Kaufman et al. eNeuron 2016, Iganaki et al., Cell 2022, Zimnik and Churchland, Nature Neuroscience 2021).

The preparatory inputs have been hypothesized to fulfill one or several (non-mutually-exclusive) possible objectives. Two notable hypotheses are that these inputs could be shaped to maximize output accuracy under regularization of the input magnitude; or that they may help the flexible re-use of the neural machinery involved in the control of movements in different contexts.

Here, the authors investigate in detail how the former hypothesis may be compatible with the presence of early inputs in recurrent network models driving arm movements, and compare models to data.

Strengths:

The authors are able to deploy an in-depth evaluation of inputs that are optimized for producing an accurate output at a pre-defined time while using a regularization term on the input magnitude, in the case of movements that are thought to be controlled in a quasi-open loop fashion such as reaches.

First, the authors have identified that optimal control theory is a great framework to study this question as it provides methods to find and analyze exact solutions to this cost function in the case of models with linear dynamics. The authors not only use this framework to get an exact assessment of how much pre-movement input arises in large recurrent networks, but also give insight into the mechanisms by which it happens by dissecting in detail low-dimensional networks. The authors find that two key network properties - observability of the readout's nullspace and limited controllability - give rise to optimal inputs that are large before the start of the movement (while the corresponding network activity lies in the nullspace of the readout). Further, the authors numerically investigate the timing of optimized inputs in models with nonlinear dynamics, and find that pre-movement inputs can also arise in these more general networks. The authors also explore how some variations on their model's constraints - such as penalizing the input roughness or changing task contingencies about the go cue timing - affect their results. Finally, the authors point out some coarse-grained similarities between the pre-movement activity driven by the optimized inputs in some of the models they studied, and the phenomenology of preparation observed in the brain during single reaches and reach sequences. Overall, the authors deploy an impressive arsenal of tools and a very in-depth analysis of their models.

Oustanding questions that could lead to interesting follow-up work:

Like all great pieces of research, this article makes it clear where current limitations lie and therefore opens up opportunities for future work.

(1) Though the optimal control theory framework is ideal for determining inputs that minimize output error while regularizing the input norm or other simple input features, it cannot easily account for some other varied types of objectives - especially those that may lead to a complex optimization landscape. For instance, the reusability of parts of the circuit, sparse use of additional neurons when learning many movements, and ease of planning (especially under uncertainty about when to start the movement), may be alternative or additional reasons that could help explain the preparatory activity observed in the brain. It is interesting to note that inputs that optimize the objective chosen by the authors arguably lead to a trade-off in terms of other desirable objectives. Specifically, the inputs the authors derive are time-dependent, so a recurrent network would be needed to produce them and it may not be easy to interpolate between them to drive new movement variants. In addition, these inputs depend on the desired time of output and therefore make it difficult to plan, e.g. in circumstances when timing should be decided depending on sensory signals. Finally, these inputs are specific to the full movement chain that will unfold, so they do not permit reuse of the inputs e.g. in movement sequences of different orders. Of note, the authors have pointed out in the discussion how their framework may be extended in future work to account for some additional objectives, such as inputs' temporal smoothness or some strategies for dealing with go cue timing uncertainty.

(2) Relatedly, if the motor circuits were to balance different types of objectives, the activity and inputs occurring before each movement may be broken down into different categories that may each specialize into their own objective. For instance, previous work (Kaufman et al. eNeuron 2016, Iganaki et al., Cell 2022, Zimnik and Churchland, Nature Neuroscience 2021) has suggested that inputs occurring before the movement could be broken down into preparatory inputs 'stricto sensu' - relating to the planned characteristics of the movement - and a trigger signal, relating to the transition from planning to execution - irrespective of whether the movement is internally timed or triggered by an external event. The current work does not address which type(s) of early input may be labeled as 'preparatory' or may be thought of as a part of 'planning' computations, or whether these inputs may come from several different source circuits. Future research could investigate these questions using a different approach, for instance, by including structural constraints from brain architecture into a neural network model.

(3) While the authors rightly point out some similarities between the inputs that they derive and observed preparatory activity in the brain, notably during motor sequences, there are also some differences. For instance, while both the derived inputs and the data show two peaks during sequences, the data reproduced from Zimnik and Churchland show preparatory inputs that have a very asymmetric shape that really plummets before the start of the next movement, whereas the derived inputs have larger amplitude during the movement period - especially for the second movement of the sequence. In addition, the data show trigger-like signals before each of the two reaches. Finally, while the data show a very high correlation between the pattern of preparatory activity of the second reach in the double reach and compound reach conditions, the derived inputs appear to be more different between the two conditions. Note that the data would be consistent with separate planning of the two reaches even in the compound reach condition, as well as the re-use of the preparatory input between the compound and double reach conditions. Therefore, different motor sequence datasets - notably, those that would show even more coarticulation between submovements - may be more promising for finding a tight match between the data and the author's inputs. In the future, further analyses in these datasets could help determine whether the coarticulation could be due to simple filtering by the circuits and muscles downstream of M1, planning of movements with adjusted curvature to mitigate the work performed by the muscles while permitting some amount of re-use across different sequences, or - as suggested by the authors - inputs fully tailored to one specific movement sequence that maximize accuracy and minimize the M1 input magnitude.

(4) Though iLQR is a powerful optimization method to find inputs optimizing the author's cost function, it also has some limitations. First, given that it relies on a linearization of the dynamics at each timestep, it has a limited ability to leverage potential advantages of nonlinearities in the dynamics. Second, the iLQR algorithm is not a biologically plausible learning rule and does not account for biological constraints affecting the circuits that produce and process these inputs. Therefore, it might be difficult for the brain to learn to produce the inputs that it finds. Consequently, when observing differences between model and data, this can confound the question of whether it comes from a difference of assumed objective or a difference of optimization procedure or circuit implementation. It remains unclear whether using alternative algorithms with different limitations - for instance, using variants of BPTT to train a separate RNN to produce the inputs in question - could impact some of the results.

(5) Under the objective considered by the authors, the amount of input occurring before the movement might be impacted by the presence of online sensory signals for closed-loop control. Even if the inputs include some sensory activity and/or the RNN activity could represent all general variables (e.g. sensory) whose states can be decoded from M1, the model does not currently include mechanisms that process imperfect (delayed, noisy) sensory feedback to adapt the output in a trial-specific manner. The information related to such sensory feedback cannot be anticipated, and therefore the related input would have to reach the motor cortex after preparation. Thus, it is an open question whether the objective and network characteristics suggested by the authors could also explain the presence of large preparatory activity before e.g. grasping movements that are thought to be more sensory-feedback-driven (Meirhaeghe et al., Cell Reports 2023).

(6) More broadly, with the type of objectives that the authors assume the inputs fulfill, some M1 properties that lead to strong preparation - notably, limited readout controllability - may not be favorable for control in general, so it would be interesting if other objectives and assumptions could robustly lead to strong preparation under more general M1 properties.'

---

## [Referee Report · Reviewer #3 (Public Review)]

I remain enthusiastic about this study. The manuscript is well-written, logical, and conceptually clear. To my knowledge, no prior modeling study has tackled the question of 'why prepare before executing, why not just execute?' Prior studies have simply assumed, to emulate empirical findings, that preparatory inputs precede execution. They never asked why. The authors show that, when there are constraints on inputs, preparation becomes a natural strategy. In contrast, with no constraint on inputs, there is no need for preparation as one could get anything one liked just via the inputs during movement. For the sake of tractability, the authors use a simple magnitude constraint: the cost function punishes the integral of the squared inputs. Thus, if small inputs before movement can reduce the size of the inputs needed during movement, preparation is a good strategy. This occurs if (and only if) the network has strong dynamics (otherwise feeding it preparatory activity would not produce anything interesting). All of this is sensible and clarifying.

As discussed in the prior round of reviews, the central constraint that the authors use is a mathematically tractable stand-in for a range of plausible (but often trickier to define and evaluate) constraints, such as simplicity of inputs (or inputs being things that other areas could provide). The manuscript now embraces this fact more explicitly and also gives some results showing that other constraints (such as on the derivative of activity, which is one component of complexity) can have the same effect. The manuscript also now discusses and addresses a modest weakness of the previous manuscript: the preparatory activity in their simulations is often overly complex temporally, lacking the (rough) plateau typically seen for data. Depending on your point of view, this is simply 'window dressing', but from my perspective it was important to know that their approach could yield more realistic-looking preparatory activity.

The most recent version of the manuscript also has a useful section in the Discussion on the topic of preparation when there is no external delay, which I found helpful given prior behavioral and physiological studies arguing that preparation can (1) be very brief, but (2) is always present. These findings mesh nicely with the authors' central result that preparation is a good network strategy, and that it would thus be normative for there to be at least a brief interval of preparation even when not imposed externally.

---

## [Author Response]

The following is the authors’ response to the previous reviews.

**Reviewer 1:**

Thank you for your review and pointing out multiple things to be discussed and clarified! Below, we go through the various limitations you pointed out and refer to the places where we have tried to address them.

(1) It's important to keep in mind that this work involves simplified models of the motor system, and often the terminology for 'motor cortex' and 'models of motor cortex' are used interchangeably, which may mislead some readers. Similarly, the introduction fails in many cases to state what model system is being discussed (e.g. line 14, line 29, line 31), even though these span humans, monkeys, mice, and simulations, which all differ in crucial ways that cannot always be lumped together.

That is a good point. We have clarified this in the text (Introduction and Discussion), to highlight the fact that our model isn’t necessarily meant to just capture M1. We have also updated the introduction to make it more clear which species the experiments which motivate our investigation were performed in.

(2) At multiple points in the manuscript thalamic inputs during movement (in mice) is used as a motivation for examining the role of preparation. However, there are other more salient motivations, such as delayed sensory feedback from the limb and vision arriving in the motor cortex, as well as ongoing control signals from other areas such as the premotor cortex.

Yes – the motivation for thalamic inputs came from the fact that those have specifically been shown to be necessary for accurate movement generation in mice. However, it is true that the inputs in our model are meant to capture any signals external to the dynamical system modeled, and as such are likely to represent a mixture of sensory signals, and feedback from other areas. We have clarified this in the Discussion, and have added this additional motivation in the Introduction.

(3) Describing the main task in this work as a delayed reaching task is not justified without caveats (by the authors' own admission: line 687), since each network is optimized with a fixed delay period length. Although this is mentioned to the reader, it's not clear enough that the dynamics observed during the delay period will not resemble those in the motor cortex for typical delayed reaching tasks.

Yes, we completely agree that the terminology might be confusing. While the task we are modeling is a delayed reaching task, it does differ from the usual setting since the network has knowledge of the delay period, and that is indeed a caveat of the model. We have added a brief paragraph just after the description of the optimal control objective to highlight this limitation.

We have also performed additional simulations using two different variants of a model-predictive control approach that allow us to relax the assumption that the go-cue time is known in advance. We show that these modifications of the optimal controller yield results that remain consistent with our main conclusions, and can in fact in some settings lead to preparatory activity plateaus during the preparation epoch as often found in monkey M1 (e.g in Elsayed et al. 2016). We have modified the Discussion to explain these results and their limitations, which are summarized in a new Supplementary Figure (S9).

(4) A number of simplifications in the model may have crucial consequences for interpretation.a) Even following the toy examples in Figure 4, all the models in Figure 5 are linear, which may limit the generalisability of the findings.

While we agree that linear models may be too simplistic, much prior analyses of M1 data suggest that it is often good enough to capture key aspects of M1 dynamics; for example, the generative model underlying jPCA is linear, and Sussillo et al. (2015) showed that the internal activity of nonlinear RNN models trained to reproduce EMG data aligned best with M1 activity when heavily regularized; in this regime, the RNN dynamics were close to linear. Nevertheless, this linearity assumption is indeed convenient from a modeling viewpoint: the optimal control problem is more easily solved for linear network dynamics and the optimal trajectories are more consistent across networks. Indeed, we had originally attempted to perform the analyses of Figure 5 in the nonlinear setting, but found that while the results were overall similar to what we report in the linear regime, iLQR was occasionally trapped into local minimal, resulting in more variable results especially for inhibition-stabilized network in the strongly connected end of the spectrum. Finally, Figure 5 is primarily meant to explore to what extent motor preparation can be predicted from basic linear control-theoretic properties of the Jacobian of the dynamics; in this regard, it made sense to work with linear RNNs (for which the Jacobian is constant).

b) Crucially, there is no delayed sensory feedback in the model from the plant. Although this simplification is in some ways a strength, this decision allows networks to avoid having to deal with delayed feedback, which is a known component of closed-loop motor control and of motor cortex inputs and will have a large impact on the control policy.

This comment resonates well with Reviewer 3's remark regarding the autonomous nature (or not) of M1 during movement. Rather than thinking of our RNN models as anatomically confined models of M1 alone, we think of them as models of the dynamics which M1 implements possibly as part of a broader network involving “inter-area loops and (at some latency) sensory feedback”, and whose state appears to be near-fully decodable from M1 activity alone. We have added a paragraph of Discussion on this important point.

(5) A key feature determining the usefulness of preparation is the direction of the readout dimension. However, all readouts had a similar structure (random Gaussian initialization). Therefore, it would be useful to have more discussion regarding how the structure of the output connectivity would affect preparation, since the motor cortex certainly does not follow this output scheme.

We agree with this limitation of our model — indeed one key message of Figure 4 is that the degree of reliance on preparatory inputs depends strongly on how the dynamics align with the readout. However, this strong dependence is somewhat specific to low-dimensional models; in higher-dimensional models (most of our paper), one expects that any random readout matrix C will pick out activity dimensions in the RNN that are sufficiently aligned with the most controllable directions of the dynamics to encourage preparation.

We did consider optimizing C away (which required differentiating through the iLQR optimizer, which is possible but very costly), but the question inevitably arises what exactly should C be optimized for, and under what constraints (e.g fixed norm or not). One possibility is to optimize C with respect to the same control objective that the control inputs are optimized for, and constrain its norm (otherwise, inputs to the M1 model, and its internal activity, could become arbitrarily small as C can grow to compensate). We performed this experiment (new Supplementary Figure S7) and obtained a similar preparation index; there was one notable difference, namely that the optimized readout modes led to greater observability compared to a random readout; thus, the same amount of “muscle energy” required for a given movement could now be produced by a smaller initial condition. In turn, this led to smaller control inputs, consistent with a lower control cost overall.

Whilst we could have systematically optimized C away, we reasoned that (i) it is computationally expensive, and (ii) the way M1 affects downstream effectors is presumably “optimized” for much richer motor tasks than simple 2D reaching, such that optimizing C for a fixed set of simple reaches could lead to misleading conclusions. We therefore decided to stick with random readouts.

Additional comments:(1) The choice of cost function seems very important. Is it? For example, penalising the square of u(t) may produce very different results than penalising the absolute value.

Yes, the choice of cost function does affect the results, at least qualitatively. The absolute value of the inputs is a challenging cost to use, as iLQR relies on a local quadratic approximation of the cost function. However, we have included additional experiments in which we penalized the squared derivative of the inputs (Supplementary Figure S8; see also our response to Reviewer 3's suggestion on this topic), and we do see differences in the qualitative behavior of the model (though the main takeaway, i.e. the reliance on preparation, continues to hold). This is now referred to and discussed in the Discussion section.

(2) In future work it would be useful to consider the role of spinal networks, which are known to contribute to preparation in some cases (e.g. Prut and Fetz, 1999).(3) The control signal magnitude is penalised, but not the output torque magnitude, which highlights the fact that control in the model is quite different from muscle control, where co-contraction would be a possibility and therefore a penalty of muscle activation would be necessary. Future work should consider the role of these differences in control policy.

Thank you for pointing us to this reference! Regarding both of these concerns, we agree that the model could be greatly improved and made more realistic in future work (another avenue for this would be to consider a more realistic biophysical model, e.g. using the MotorNet library). We hope that the current Discussion, which highlights the various limitations of our modeling choices, makes it clear that a lot of these choices could easily be modified depending on the specific assumptions/investigation being performed.

**Reviewer 2:**

Thank you for your positive review! We very much agree with the limitations you pointed out, some of which overlapped with the comments of the other reviewers. We have done our best to address them through additional discussion and new supplementary figures. We briefly highlight below where those changes can be found.

(1) Though the optimal control theory framework is ideal to determine inputs that minimize output error while regularizing the input norm, it however cannot easily account for some other varied types of objectives especially those that may lead to a complex optimization landscape. For instance, the reusability of parts of the circuit, sparse use of additional neurons when learning many movements, and ease of planning (especially under uncertainty about when to start the movement), may be alternative or additional reasons that could help explain the preparatory activity observed in the brain. It is interesting to note that inputs that optimize the objective chosen by the authors arguably lead to a trade-off in terms of other desirable objectives. Specifically, the inputs the authors derive are time-dependent, so a recurrent network would be needed to produce them and it may not be easy to interpolate between them to drive new movement variants. In addition, these inputs depend on the desired time of output and therefore make it difficult to plan, e.g. in circumstances when timing should be decided depending on sensory signals. Finally, these inputs are specific to the full movement chain that will unfold, so they do not permit reuse of the inputs e.g. in movement sequences of different orders.

Yes, that is a good point! We have incorporated further Discussion related to this point. We have additionally included a new example in which we regularize the temporal complexity of the inputs (see also our response to Reviewer 3's suggestion on this topic), which leads to more slowly varying inputs, and may indeed represent a more realistic constraint and lead to simpler inputs that can more easily be interpolated between. We also agree that uncertainty about the upcoming go cue may play an important role in the strategy adopted by the animals. While we have not performed an extensive investigation of the topic, we have included a Supplementary Figure (S9) in which we used Model Predictive Control to investigate the effect of planning under uncertainty about the go cue arrival time. We hope that this will give the reader a better sense of what sort of model extensions are possible within our framework.

(2) Relatedly, if the motor circuits were to balance different types of objectives, the activity and inputs occurring before each movement may be broken down into different categories that may each specialize into one objective. For instance, previous work (Kaufman et al. eNeuron 2016, Iganaki et al., Cell 2022, Zimnik and Churchland, Nature Neuroscience 2021) has suggested that inputs occurring before the movement could be broken down into preparatory inputs 'stricto sensu' - relating to the planned characteristics of the movement - and a trigger signal, relating to the transition from planning to execution - irrespective of whether the movement is internally timed or triggered by an external event. The current work does not address which type(s) of early input may be labeled as 'preparatory' or may be thought of as a part of 'planning' computations.

Yes, our model does indeed treat inputs in a very general way, and does not distinguish between the different types of processes they may be composed of. This is partly because we do not explicitly model where the inputs come from, such that our inputs likely englobe multiple processes. We have added discussion related to this point.

(3) While the authors rightly point out some similarities between the inputs that they derive and observed preparatory activity in the brain, notably during motor sequences, there are also some differences. For instance, while both the derived inputs and the data show two peaks during sequences, the data reproduced from Zimnik and Churchland show preparatory inputs that have a very asymmetric shape that really plummets before the start of the next movement, whereas the derived inputs have larger amplitude during the movement period - especially for the second movement of the sequence. In addition, the data show trigger-like signals before each of the two reaches. Finally, while the data show a very high correlation between the pattern of preparatory activity of the second reach in the double reach and compound reach conditions, the derived inputs appear to be more different between the two conditions. Note that the data would be consistent with separate planning of the two reaches even in the compound reach condition, as well as the re-use of the preparatory input between the compound and double reach conditions. Therefore, different motor sequence datasets - notably, those that would show even more coarticulation between submovements - may be more promising to find a tight match between the data and the author's inputs. Further analyses in these datasets could help determine whether the coarticulation could be due to simple filtering by the circuits and muscles downstream of M1, planning of movements with adjusted curvature to mitigate the work performed by the muscles while permitting some amount of re-use across different sequences, or - as suggested by the authors - inputs fully tailored to one specific movement sequence that maximize accuracy and minimize the M1 input magnitude.

Regarding the exact shape of the occupancy plots, it is important to note that some of the more qualitative aspects (e.g the relative height of the two peaks) will change if we change the parameters of the cost function. Right now, we have chosen the parameters to ensure that both reaches would be performed at roughly the same speed (as a way to very loosely constrain the parameters based on the observed behavior). However, small changes to the hyperparameters can lead to changes in the model output (e.g one of the two consecutive reaches being performed using greater acceleration than the other), and since our biophysical model is fairly simple, changes in the behavior are directly reflected in the network activity. Essentially, what this means is that while the double occupancy is a consistent feature of the model, the exact shape of the peaks is more sensitive to hyperparameters, and we do not wish to draw any strong conclusions from them, given the simplicity of the biophysical model. However, we do agree that our model exhibits some differences with the data. As discussed above, we have included additional discussion regarding the potential existence of separate inputs for planning vs triggering the movement in the context of single reaches.

Overall, we are excited about the suggestions made by the Reviewer here about using our approach to analyze other motor sequence datasets, but we think that in order to do this properly, one would need to adopt a more realistic musculo-skeletal model (such as one provided by MotorNet).

(4) Though iLQR is a powerful optimization method to find inputs optimizing the author's cost function, it also has some limitations. First, given that it relies on a linearization of the dynamics at each timestep, it has a limited ability to leverage potential advantages of nonlinearities in the dynamics. Second, the iLQR algorithm is not a biologically plausible learning rule and therefore it might be difficult for the brain to learn to produce the inputs that it finds. It remains unclear whether using alternative algorithms with different limitations - for instance, using variants of BPTT to train a separate RNN to produce the inputs in question - could impact some of the results.

We agree that our choice of iLQR has limitations: while it offers the advantage of convergence guarantees, it does indeed restrict the choice of cost function and dynamics that we can use. We have now included extensive discussion of how the modeling choices affect our results.

We do not view the lack of biological plausibility of iLQR as an issue, as the results are agnostic to the algorithm used for optimization. However, we agree that any structure imposed on the inputs (e.g by enforcing them to be the output of a self-contained dynamical system) would likely alter the results. A potentially interesting extension of our model would be to do just what the reviewer suggested, and try to learn a network that can generate the optimal inputs. However, this is outside the scope of our investigation, as it would then lead to new questions (e.g what brain region would that other RNN represent?).

(5) Under the objective considered by the authors, the amount of input occurring before the movement might be impacted by the presence of online sensory signals for closed-loop control. It is therefore an open question whether the objective and network characteristics suggested by the authors could also explain the presence of preparatory activity before e.g. grasping movements that are thought to be more sensory-driven (Meirhaeghe et al., Cell Reports 2023).

It is true that we aren’t currently modeling sensory signals explicitly. However, some of the optimal inputs we infer may be capturing upstream information which could englobe some sensory information. This is currently unclear, and would likely depend on how exactly the model is specified. We have added new discussion to emphasize that our dynamics should not be understood as just representing M1, but more general circuits whose state can be decoded from M1.

**Reviewer #2 (Recommendations For The Authors):**

Additionally, thank you for pointing out various typos in the manuscript, we have fixed those!

**Reviewer 3:**

Thank you very much for your review, which makes a lot of very insightful points, and raises several interesting questions. In summary, we very much agree with the limitations you pointed out. In particular, the choice of input cost is something we had previously discussed, but we had found it challenging to decide on what a reasonable cost for “complexity” could be. Following your comment, we have however added a first attempt at penalizing “temporal complexity”, which shows promising behavior. We have only included those additional analyses as supplementary figures, and we have included new discussion, which hopefully highlights what we meant by the different model components, and how the model behavior may change as we vary some of our choices. We hope this can be informative for future models that may use a similar approach. Below, we highlight the changes that we have made to address your comments.

The main limitation of the study is that it focuses exclusively on one specific constraint - magnitude - that could limit motor-cortex inputs. This isn't unreasonable, but other constraints are at least as likely, if less mathematically tractable. The basic results of this study will probably be robust with regard such issues - generally speaking, any constraint on what can be delivered during execution will favor the strategy of preparing - but this robustness cuts both ways. It isn't clear that the constraint used in the present study - minimizing upstream energy costs - is the one that really matters. Upstream areas are likely to be limited in a variety of ways, including the complexity of inputs they can deliver. Indeed, one generally assumes that there are things that motor cortex can do that upstream areas can't do, which is where the real limitations should come from. Yet in the interest of a tractable cost function, the authors have built a system where motor cortex actually doesn't do anything that couldn't be done equally well by its inputs. The system might actually be better off if motor cortex were removed. About the only thing that motor cortex appears to contribute is some amplification, which is 'good' from the standpoint of the cost function (inputs can be smaller) but hardly satisfying from a scientific standpoint.The use of a term that punishes the squared magnitude of control signals has a long history, both because it creates mathematical tractability and because it (somewhat) maps onto the idea that one should minimize the energy expended by muscles and the possibility of damaging them with large inputs. One could make a case that those things apply to neural activity as well, and while that isn't unreasonable, it is far from clear whether this is actually true (and if it were, why punish the square if you are concerned about ATP expenditure?). Even if neural activity magnitude an important cost, any costs should pertain not just to inputs but to motor cortex activity itself. I don't think the authors really wish to propose that squared input magnitude is the key thing to be regularized. Instead, this is simply an easily imposed constraint that is tractable and acts as a stand-in for other forms of regularization / other types of constraints. Put differently, if one could write down the 'true' cost function, it might contain a term related to squared magnitude, but other regularizing terms would by very likely to dominate. Using only squared magnitude is a reasonable way to get started, but there are also ways in which it appears to be limiting the results (see below).I would suggest that the study explore this topic a bit. Is it possible to use other forms of regularization? One appealing option is to constrain the complexity of inputs; a long-standing idea is that the role of motor cortex is to take relatively simple inputs and convert them to complex time-evolving inputs suitable for driving outputs. I realize that exploring this idea is not necessarily trivial. The right cost-function term is not clear (should it relate to low-dimensionality across conditions, or to smoothness across time?) and even if it were, it might not produce a convex cost function. Yet while exploring this possibility might be difficult, I think it is important for two reasons.First, this study is an elegant exploration of how preparation emerges due to constraints on inputs, but at present that exploration focuses exclusively on one constraint. Second, at present there are a variety of aspects of the model responses that appear somewhat unrealistic. I suspect most of these flow from the fact that while the magnitude of inputs is constrained, their complexity is not (they can control every motor cortex neuron at both low and high frequencies). Because inputs are not complexity-constrained, preparatory activity appears overly complex and never 'settles' into the plateaus that one often sees in data. To be fair, even in data these plateaus are often imperfect, but they are still a very noticeable feature in the response of many neurons. Furthermore, the top PCs usually contain a nice plateau. Yet we never get to see this in the present study. In part this is because the authors never simulate the situation of an unpredictable delay (more on this below) but it also seems to be because preparatory inputs are themselves strongly time-varying. More realistic forms of regularization would likely remedy this.

That is a very good point, and it mirrors several concerns that we had in the past. While we did focus on the input norm for the sake of simplicity, and because it represents a very natural way to regularize our control solutions, we agree that a “complexity cost” may be better suited to models of brain circuits. We have addressed this in a supplementary investigation. We chose to focus on a cost that penalizes the temporal complexity of the inputs, as ||u(t+1) - u(t)||^2. Note that this required augmenting the state of the model, making the computations quite a bit slower; while it is doable if we only penalize the first temporal derivative, it would not scale well to higher orders.

Interestingly, we did find that the activity in that setting was somewhat more realistic (see new Supplementary Figure S8), with more sustained inputs and plateauing activity. While we have kept the original model for most of the investigations, the somewhat more realistic nature of the results under that setting suggests that further exploration of penalties of that sort could represent a promising avenue to improve the model.

We also found the idea of a cost that would ensure low-dimensionality of the inputs across conditions very interesting. However, it is challenging to investigate with iLQR as we perform the optimization separately for each condition; nevertheless, it could be investigated using a different optimizer.

At present, it is also not clear whether preparation always occurs even with no delay. Given only magnitude-based regularization, it wouldn't necessarily have to be. The authors should perform a subspace-based analysis like that in Figure 6, but for different delay durations. I think it is critical to explore whether the model, like monkeys, uses preparation even for zero-delay trials. At present it might or might not. If not, it may be because of the lack of more realistic constraints on inputs. One might then either need to include more realistic constraints to induce zero-delay preparation, or propose that the brain basically never uses a zero delay (it always delays the internal go cue after the preparatory inputs) and that this is a mechanism separate from that being modeled.I agree with the authors that the present version of the model, where optimization knows the exact time of movement onset, produces a reasonably realistic timecourse of preparation when compared to data from self-paced movements. At the same time, most readers will want to see that the model can produce realistic looking preparatory activity when presented with an unpredictable delay. I realize this may be an optimization nightmare, but there are probably ways to trick the model into optimizing to move soon, but then forcing it to wait (which is actually what monkeys are probably doing). Doing so would allow the model to produce preparation under the circumstances where most studies have examined it. In some ways this is just window-dressing (showing people something in a format they are used to and can digest) but it is actually more than that, because it would show that the model can produce a reasonable plateau of sustained preparation. At present it isn't clear it can do this, for the reasons noted above. If it can't, regularizing complexity might help (and even if this can't be shown, it could be discussed).In summary, I found this to be a very strong study overall, with a conceptually timely message that was well-explained and nicely documented by thorough simulations. I think it is critical to perform the test, noted above, of examining preparatory subspace activity across a range of delay durations (including zero) to see whether preparation endures as it does empirically. I think the issue of a more realistic cost function is also important, both in terms of the conceptual message and in terms of inducing the model to produce more realistic activity. Conceptually it matters because I don't think the central message should be 'preparation reduces upstream ATP usage by allowing motor cortex to be an amplifier'. I think the central message the authors wish to convey is that constraints on inputs make preparation a good strategy. Many of those constraints likely relate to the fact that upstream areas can't do things that motor cortex can do (else you wouldn't need a motor cortex) and it would be good if regularization reflected that assumption. Furthermore, additional forms of regularization would likely improve the realism of model responses, in ways that matter both aesthetically and conceptually. Yet while I think this is an important issue, it is also a deep and tricky one, and I think the authors need considerable leeway in how they address it. Many of the cost-function terms one might want to use may be intractable. The authors may have to do what makes sense given technical limitations. If some things can't be done technically, they may need to be addressed in words or via some other sort of non-optimization-based simulation.Specific commentsAs noted above, it would be good to show that preparatory subspace activity occurs similarly across delay durations. It actually might not, at present. For a zero ms delay, the simple magnitude-based regularization may be insufficient to induce preparation. If so, then the authors would either have to argue that a zero delay is actually never used internally (which is a reasonable argument) or show that other forms of regularization can induce zero-delay preparation.

Yes, that is a very interesting analysis to perform, which we had not considered before! When investigating this, we found that the zero-delay strategy does not rely on preparation in the same way as is seen in the monkeys. This seems to be a reflection of the fact that our “Go cue” corresponds to an “internal” go cue which would likely come after the true, “external go cue” – such that we would indeed never actually be in the zero delay setting. This is not something we had addressed (or really considered) before, although we had tried to ensure we referred to “delta prep” as the duration of the preparatory period but not necessarily the delay period. We have now included more discussion on this topic, as well as a new Supplementary Figure S10.

I agree with the authors that prior modeling work was limited by assuming the inputs to M1, which meant that prior work couldn't address the deep issue (tackled here) of why there should be any preparatory inputs at all. At the same time, the ability to hand-select inputs did provide some advantages. A strong assumption of prior work is that the inputs are 'simple', such that motor cortex must perform meaningful computations to convert them to outputs. This matters because if inputs can be anything, then they can just be the final outputs themselves, and motor cortex would have no job to do. Thus, prior work tried to assume the simplest inputs possible to motor cortex that could still explain the data. Most likely this went too far in the 'simple' direction, yet aspects of the simplicity were important for endowing responses with realistic properties. One such property is a large condition-invariant response just before movement onset. This is a very robust aspect of the data, and is explained by the assumption of a simple trigger signal that conveys information about when to move but is otherwise invariant to condition. Note that this is an implicit form of regularization, and one very different from that used in the present study: the input is allowed to be large, but constrained to be simple. Preparatory inputs are similarly constrained to be simple in the sense that they carry only information about which condition should be executed, but otherwise have little temporal structure. Arguably this produces slightly too simple preparatory-period responses, but the present study appears to go too far in the opposite direction. I would suggest that the authors do what they can to address these issue via simulations and/or discussion. I think it is fine if the conclusion is that there exist many constraints that tend to favor preparation, and that regularizing magnitude is just one easy way of demonstrating that. Ideally, other constraints would be explored. But even if they can't be, there should be some discussion of what is missing - preparatory plateaus, a realistic condition-invariant signal tied to movement onset - under the present modeling assumptions.

As described above, we have now included two additional figures. In the first one (S8, already discussed above), we used a temporal smoothness prior, and we indeed get slightly more realistic activity plateaus. In a second supplementary figure (S9), we have also considered using model predictive control (MPC) to optimize the inputs under an uncertain go cue arrival time. There, we found that removing the assumption that the delay period is known came with new challenges: in particular, it requires the specification of a “mental model” of when the Go cue will arrive. While it is reasonable to expect that monkeys will have a prior over the go time arrival cue that will be shaped by the design of the experiment, some assumptions must be made about the utility functions that should be used to weigh this prior. For instance, if we imagine that monkeys carry a model of the possible arrival time of the go cue that is updated online, they could nonetheless act differently based on this information, for instance by either preparing so as to be ready for the earliest go cue possible or alternatively to be ready for the average go cue. This will likely depend on the exact task design and reward/penalty structure. Here, we added simulations with those two cases (making simplifying assumptions to make the problem tractable/solvable using model predictive control), and found that the “earliest preparation” strategy gives rise to more realistic plateauing activity, while the model where planning is done for the “most likely go time” does not. We suspect that more realistic activity patterns could be obtained by e.g combining this framework with the temporal smoothness cost. However, the main point we wished to make with this new supplementary figure is that it is possible to model the task in a slightly more realistic way (although here it comes at the cost of additional model assumptions). We have now added more discussion related to those points. Note that we have kept our analyses on these new models to a minimum, as the main takeaway we wish to convey from them is that most components of the model could be modified/made more realistic. This would impact the qualitative behavior of the system and match to data but – in the examples we have so far considered – does not appear to modify the general strategy of networks relying on preparation.

On line 161, and in a few other places, the authors cite prior work as arguing for "autonomous internal dynamics in M1". I think it is worth being careful here because most of that work specifically stated that the dynamics are likely not internal to M1, and presumably involve inter-area loops and (at some latency) sensory feedback. The real claim of such work is that one can observe most of the key state variables in M1, such that there are periods of time where the dynamics are reasonably approximated as autonomous from a mathematical standpoint. This means that you can estimate the state from M1, and then there is some function that predicts the future state. This formal definition of autonomous shouldn't be conflated with an anatomical definition.

Yes, that is a good point, thank you for making it so clearly! Indeed, as previous work, we do not think of our “M1 dynamics” as being internal to M1, but they may instead include sensory feedback / inter-area loops, which we summarize into the connectivity, that we chose to have dynamics that qualitatively resemble data. We have now incorporated more discussion regarding what exactly the dynamics in our model represent.

Round 2 of reviews

**Reviewer 3:**
My remaining comments largely pertain to some subtle (but to me important) nuances at a few locations in the text. These should be easy for the authors to address, in whatever way they see fit.Specific comments:(1) The authors state the following on line 56: "For preparatory processes to avoid triggering premature movement, any pre-movement activity in the motor and dorsal pre-motor (PMd) cortices must carefully exclude those pyramidal tract neurons."This constraint is overly restrictive. PT neurons absolutely can change their activity during preparation in principle (and appear to do so in practice). The key constraint is looser: those changes should have no net effect on the muscles. E.g., if d is the vector of changes in PT neuron firing rates, and b is the vector of weights, then the constraint is that b'd = 0. d = 0 is one good way of doing this, but only one. Half the d's could go up and half could go down. Or they all go up, but half the b's are negative. Put differently, there is no reason the null space has to be upstream of the PT neurons. It could be partly, or entirely, downstream. In the end, this doesn't change the point the authors are making. It is still the case that d has to be structured to avoid causing muscle activity, which raises exactly the point the authors care about: why risk this unless preparation brings benefits? However, this point can be made with a more accurate motivation. This matters, because people often think that a null-space is a tricky thing to engineer, when really it is quite natural. With enough neurons, preparing in the null space is quite simple.

That is a good point – we have now reformulated this sentence to instead say “to avoid triggering premature movement, any pre-movement activity in the motor and dorsal premotor (PMd) cortices must engage the pyramidal tract neurons in a way that ensures their activity patterns will not lead to any movement”.

(2) Line 167: 'near-autonomous internal dynamics in M1'.It would be good if such statements, early in the paper, could be modified to reflect the fact that the dynamics observed in M1 may depend on recurrence that is NOT purely internal to M1. A better phrase might be 'near-autonomous dynamics that can be observed in M1'. A similar point applies on line 13. This issue is handled very thoughtfully in the Discussion, starting on line 713. Obviously it is not sensible to also add multiple sentences making the same point early on. However, it is still worth phrasing things carefully, otherwise the reader may have the wrong impression up until the Discussion (i.e. they may think that both the authors, and prior studies, believe that all the relevant dynamics are internal to M1). If possible, it might also be worth adding one sentence, somewhere early, to keep readers from falling into this hole (and then being stuck there till the Discussion digs them out).

That is a good point: we have now edited the text after line 170 to make it clear that the underlying dynamics may not be confined to M1, and have referenced the later discussion there.

(3) The authors make the point, starting on line 815, that transient (but strong) preparatory activity empirically occurs without a delay. They note that their model will do this but only if 'no delay' means 'no external delay'. For their model to prepare, there still needs to be an internal delay between when the first inputs arrive and when movement generating inputs arrive.This is not only a reasonable assumption, but is something that does indeed occur empirically. This can be seen in Figure 8c of Lara et al. Similarly, Kaufman et al. 2016 noted that "the sudden change in the CIS [the movement triggering event] occurred well after (~150 ms) the visual go cue... (~60 ms latency)" Behavioral experiments have also argued that internal movement-triggering events tend to be quite sluggish relative to the earliest they could be, causing RTs to be longer than they should be (Haith et al. Independence of Movement Preparation and Movement Initiation). Given this empirical support, the authors might wish to add a sentence indicating that the data tend to justify their assumption that the internal delay (separating the earliest response to sensory events from the events that actually cause movement to begin) never shrinks to zero.While on this topic, the Haith and Krakauer paper mentioned above good to cite because it does ponder the question of whether preparation is really necessary. By showing that they could get RTs to shrink considerably before behavior became inaccurate, they showed that people normally (when not pressured) use more preparation time than they really need. Given Lara et al, we know that preparation does always occur, but Haith and Krakauer were quite right that it can be very brief. This helped -- along with neural results -- change our view of preparation from something more cognitive that had to occur, so something more mechanical that was simply a good network strategy, which is indeed the authors current point. Working a discussion of this into the current paper may or may not make sense, but if there is a place where it is easy to cite, it would be appropriate.

This is a nice suggestion, and we thank the reviewer for pointing us to the Haith and Krakauer paper. We have now added this reference and extended the paragraph following line 815 to briefly discuss the possible decoupling between preparation and movement initiation that is shown in the Haith paper, emphasizing how this may affect the interpretation of the internal delay and comparisons with behavioral experiments.